# Merging Memory and Space: A State Space Neural Operator

**Nodens F. Koren**                                     *nodens.koren@inf.ethz.ch*
*Department of Computer Science*
*ETH Zürich*

**Samuel Lanthaler**                                *samuel.lanthaler@univie.ac.at*
*Faculty of Mathematics*
*University of Vienna*

**Reviewed on OpenReview:** https://openreview.net/forum?id=SwLxxz0x58

## Abstract

We propose the *State Space Neural Operator* (SS-NO), a compact architecture for learning solution operators of time-dependent partial differential equations (PDEs). Our formulation extends structured state space models (SSMs) to joint spatiotemporal modeling, introducing two key mechanisms: *adaptive damping*, which stabilizes learning by localizing receptive fields, and *learnable frequency modulation*, which enables data-driven spectral selection. These components provide a unified framework for capturing long-range dependencies with parameter efficiency. Theoretically, we establish connections between SSMs and neural operators, proving a universality theorem for convolutional architectures with a full field of view. Empirically, SS-NO achieves strong performance across diverse PDE benchmarks—including 1D Burgers' and Kuramoto–Sivashinsky equations, and 2D Navier–Stokes and compressible Euler flows—while using significantly fewer parameters than competing approaches. Our results demonstrate that state space modeling provides an effective foundation for efficient and accurate neural operator learning.

## 1 Introduction

Many problems in scientific computing require approximating nonlinear operators that map input functions to output functions, often governed by partial differential equations (PDEs). Neural operators (NOs) (Kovachki et al., 2023) provide a mesh-independent framework for this task, operating directly on functions and generalizing across discretizations.

The Fourier Neural Operator (FNO) (Li et al., 2021) is a widely used NO, as its global convolution kernels effectively capture long-range spatial correlations. However, this fully global design comes with substantial computational and memory costs, scaling poorly in higher dimensions. Factorized variants (Tran et al., 2023; Lehmann et al., 2023; 2024) address this by decomposing operator learning into lower-dimensional subspaces, achieving linear scaling in dimension—a crucial advantage for high-dimensional spatial applications. Yet, these approaches offer limited flexibility in adjusting receptive fields or frequency modes to different spatial regions or temporal dynamics.

In a complementary line of work, state space models (SSMs) (Gu et al., 2022b), such as the diagonalized S4D architecture (Gu et al., 2022a), have achieved remarkable success in modeling long-range *temporal* dependencies. Like factorized FNOs, SSMs scale linearly in time and memory, but they provide an additional advantage: data-dependent parameterization of convolutional filters that can learn to emphasize different frequency components and temporal scales. Despite these parallels, the connection between FNOs (primarily spatial) and SSMs (primarily temporal) has remained underexplored.

This paper makes that connection explicit. We introduce the **State Space Neural Operator (SS-NO)**, a unified operator learning framework that applies SSMs directly over joint spatiotemporal domains. SS-NO

can be interpreted in two complementary ways: as extending SSMs from purely temporal modeling to spatiotemporal operator learning, or as generalizing FNOs into more expressive architectures with adaptive receptive fields, frequency content, and temporal causality.

Theoretically, we prove that any continuous operator can be approximated arbitrarily well by convolutional NOs with full receptive fields, a condition satisfied by both FNO and SS-NO. Importantly, the spatial-only variant of SS-NO subsumes factorized FNO as a special case while adding two key capabilities: learned damping coefficients for stability control and data-driven frequency modulation. Empirically, we validate SS-NO on challenging 1D and 2D benchmarks, including chaotic Kuramoto–Sivashinsky dynamics, variants of Kolmogorov flow, Richtmyer–Meshkov instability, and gravitational Rayleigh–Taylor turbulence. Across tasks, SS-NO surpasses existing methods using fewer parameters. These results highlight SS-NO as a principled, scalable, and practical architecture for data-driven modeling in engineering and the physical sciences.

## 2 Related Work

### 2.1 Neural Operators

Data-driven neural operators have emerged as a powerful framework for learning PDE solution operators (Chen & Chen, 1995; Bhattacharya et al., 2021; Lu et al., 2021; Kovachki et al., 2023). Theoretical work established that several neural operators can universally approximate nonlinear operators (Chen & Chen, 1995; Kovachki et al., 2021; 2024; Lanthaler et al., 2024), including DeepONet (Lu et al., 2021; Lanthaler et al., 2022) and the Fourier Neural Operator (FNO), which implements global convolution via the Fourier transform (Li et al., 2021). Extensions such as the Factorized Fourier Neural Operator (FFNO) (Tran et al., 2023) enhance efficiency and expressivity in practice, but their theoretical expressivity remains unknown. Alternative inductive biases have also been explored, including U-Net–based convolutional operators (Raonic et al., 2023; Gupta & Brandstetter, 2023; Rahman et al., 2023; Takamoto et al., 2022) and attention-based operators like the Galerkin Transformer (GKT) (Cao, 2021) and the FactFormer (Li et al., 2023a).

While some studies suggest that purely Markovian models can be performant for learning time-evolution PDEs (Tran et al., 2023; Lippe et al., 2023), recent work has shown that incorporating past states improves accuracy, particularly under low-resolution or noisy conditions. This has led to the integration of memory mechanisms into neural operators. Buitrago et al. (2025) introduce the Memory Neural Operator (MemNO) framework, which embeds temporal S4-based recurrence within general neural operator architectures, motivated by the Mori–Zwanzig formalism (Mori, 1965; Zwanzig, 1961). Our work extends this direction by examining scenarios with missing contextual information and introducing a spatiotemporal factorization that distributes memory across *both time and space.*

### 2.2 Structured State Space Models (SSMs)

Structured state space models (SSMs) have recently achieved strong results on long-range sequence tasks in natural language processing and vision. The Structured State Space sequence model (S4) (Gu et al., 2022b;a) uses a continuous-time linear SSM layer to capture very long-range dependencies efficiently, and the Mamba model (Gu & Dao, 2024) extends this idea by making the SSM parameters input-dependent, further improving performance on large-scale language modeling tasks. These SSM-based architectures have been shown to match or exceed Transformer performance on a variety of benchmark tasks.

SSMs have also begun to be used in operator learning. For example, Hu et al. (2024) incorporate Mamba-style SSM layers to predict dynamical systems efficiently. In the context of neural PDE solvers, recent works like Cheng et al. (2024) and Zheng et al. (2024) embed Mamba SSM modules to capture spatial correlations in solution fields. These methods primarily focus on spatial modeling, applying SSM layers across spatial dimensions or treating time steps sequentially. By contrast, our work focuses on applying SSMs jointly in both space and time, employing a unified spatiotemporal SSM architecture for neural operator learning.

## 3 Methodology

### 3.1 Problem Formulation

Let $\Omega \subset \mathbb{R}^d$ be a bounded spatial domain, and consider the solution $u \in C([0,T]; L^2(\Omega; \mathbb{R}^V))$ of a time-dependent partial differential equation (PDE), where $C$ denotes the space of continuous functions and $L^2$ denotes the space of square-integrable functions. We assume access to a dataset of $N$ solution trajectories $u^{(i)}(t,x)_{i=1}^N$ generated from varying initial or parametric conditions.

In practice, we discretize the spatial domain $\Omega$ using an equispaced grid $\mathcal{S}$ of resolution $f$, and the temporal domain $[0,T]$ using an equispaced grid $\mathcal{T}$ with $N_t + 1$ points. Let $T_{\text{in}} < T$ denote the fixed input horizon.

Given the trajectory segment $u(t,x)|_{t \in [0,T_{\text{in}}]}$ on $\mathcal{S}$, our goal is to predict the future evolution $u(t,x)|_{t \in [T_{\text{in}},T]}$ on the same grid. Rather than forecasting a single step, we aim to learn the full system dynamics conditioned on the initial segment. Formally, the NO model defines a parametric map $\Psi_\theta : C(\mathcal{T}_{\text{in}}; L^2(\Omega; \mathbb{R}^V)) \to C(\mathcal{T}_{\text{out}}; L^2(\Omega; \mathbb{R}^V))$, where $\mathcal{T}_{\text{in}} = [0, T_{\text{in}}]$ and $\mathcal{T}_{\text{out}} = [T_{\text{in}}, T]$, mapping the input segment to the predicted solution.

### 3.2 Structured State Space Models (S4 and S4D)

We begin with the continuous-time linear state space model (SSM) defined as:

$$\frac{d}{dt}v(t) = Av(t) + Bu(t),$$
$$y(t) = Cv(t) + Du(t), \tag{1}$$

where $v(t) \in \mathbb{R}^H$ is the hidden state, $H$ is the number of states, $u(t) \in \mathbb{R}$ is the input, and $y(t) \in \mathbb{R}$ is the output. The matrices $A \in \mathbb{R}^{H \times H}$, $B \in \mathbb{R}^{H \times 1}$, $C \in \mathbb{R}^{1 \times H}$, and $D \in \mathbb{R}$ define the system dynamics.

The Structured State Space (S4) model (Gu et al., 2022b) introduces a parameterization where $A$ is structured to allow efficient computation of the convolution kernel $\kappa(t)$ corresponding to the system's impulse response. Specifically, S4 utilizes a diagonal plus low-rank (DPLR) structure, $A = \Lambda + PQ^\top$, where $\Lambda \in \mathbb{C}^{H \times H}$ is diagonal, and $P, Q \in \mathbb{C}^{H \times r}$ with $r \ll H$. This structure enables computation of $\kappa(t)$ via the Cauchy kernel and allows for efficient implementation with the Fast Fourier Transform (FFT).

To further simplify the model, S4D (Gu et al., 2022a) restricts $A$ to be diagonal, i.e., $A = \text{diag}(\lambda_1, \ldots, \lambda_N)$, and initializes the eigenvalues $\{\lambda_i\}$ to approximate the behavior of the original S4 model. This diagonalization reduces the computational complexity and memory footprint while retaining the ability to model long-range dependencies.

In both S4 and S4D, after computing the convolution with the kernel $\kappa(t)$, a pointwise nonlinearity $\sigma(\cdot)$ (e.g., GELU) is applied, followed by a residual connection.

### 3.3 Markovian Spatial State Space Model

To model spatial dependencies in PDEs, we extend the S4D framework to spatial dimensions. A key requirement for universality of factorized convolutional neural operators is a full field of view, as formalized in Theorem 4.1. A unidirectional SSM does not satisfy this condition because each spatial location only aggregates information from one side, limiting expressivity. Empirically, we confirm in Appendix C.1 that unidirectional scans underperform, motivating the use of bidirectional processing.

For a 1D spatial domain discretized into $X$ points, we define two SSM models: $\mathcal{M}_{fwd}$ and $\mathcal{M}_{bwd}$, where the $_{fwd}$ and $_{bwd}$ labels serve as identifiers rather than indicating scan direction. The processing is computed as

$$y_{fwd} = \mathcal{M}_{fwd}(u), \qquad y_{bwd} = \text{flip}\left(\mathcal{M}_{bwd}(\text{flip}(u))\right), \tag{2}$$

with the final output $y = y_{fwd} + y_{bwd}$. For multi-dimensional grids, such as $X \times Y$, bidirectional SSMs are applied sequentially along each axis (first $x$, then $y$), ensuring every spatial location has a full field of view and satisfying Theorem 4.1.

**Comparison with Existing Methods.** Unlike Vision Mamba (Zhu et al., 2024), which processes 2D data in a zigzag manner using bidirectional SSMs, our approach applies SSMs separately along each spatial dimension. While Factorized Fourier Neural Operators (F-FNO) (Tran et al., 2023) process each spatial dimension in parallel and sum the results, our method applies SSMs sequentially, allowing more expressive modeling of spatial interactions. Due to memory constraints, we do not employ Mamba-based SSMs directly but explore alternative 2D and factorized spatial SSM configurations using S4D in Appendix D.

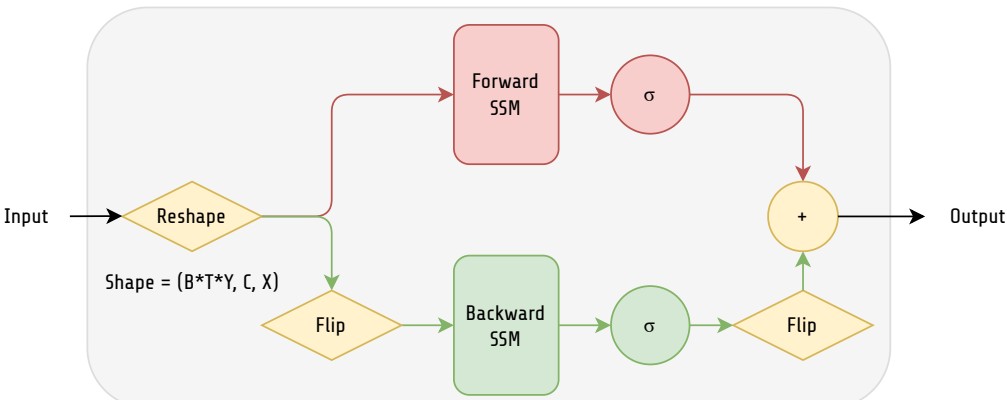

Figure 1: Detailed illustration of the spatial bidirectional SSM module. $B$: batch size, $T$: temporal length, $X$: spatial dimensions, $C$: input channels, $\sigma$: pointwise nonlinearity, and $+$: element-wise addition. The input is processed through both a forward spatial SSM and a flipped backward spatial SSM. Each path includes a residual connection and nonlinear activation, and their outputs are aggregated to form the final output.

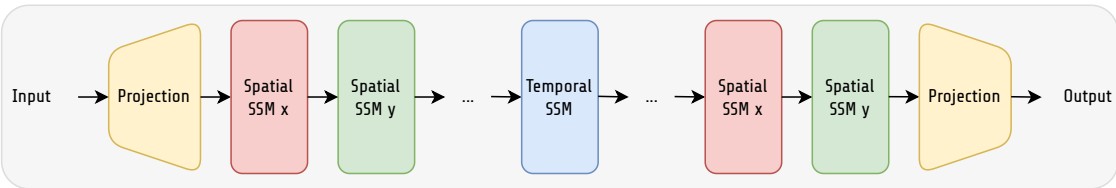

Figure 2: Architecture combining Markovian 1D spatial SSM modules with a single temporal SSM following the MemNO framework. The spatial SSMs are applied sequentially across spatial dimensions, while the temporal SSM's position within the stack is a tunable hyperparameter.

### 3.4 Full SS-NO Model with Temporal Memory

The SS-NO architecture combines Markovian spatial SSM layers (Section 3.3) with a single non-Markovian temporal memory module following the MemNO framework to capture spatiotemporal dependencies in PDEs. Spatial layers process local interactions sequentially across dimensions, while the temporal module aggregates information across previous timesteps. During inference, the temporal module maintains and updates a hidden state at each step, which is combined with spatial processing for next-step predictions.

The number of spatial layers and the placement of the temporal layer within the network stack are tunable hyperparameters optimized for specific PDE characteristics. Figures 1 and 2 illustrate the architecture: spatial SSM modules perform bidirectional processing with residual connections, while the full model integrates these with the temporal module. Complete implementation details are provided in Appendix G.

## 4 Theory

We view the proposed SS-NO architecture as an instance of a popular neural operator paradigm Kovachki et al. (2023), combining two types of layers: (a) pointwise composition with nonlinear activations, and (b) application of nonlocal convolution operators, and with prototypical hidden layers of the form,

$$\mathcal{L}: \ v(x) \mapsto \underbrace{\sigma(Wv(x) + b)}_{\text{(a) nonlinear}} + \underbrace{\int_D \kappa(x - y)v(y)\,dy}_{\text{(b) nonlocal}}), \tag{3}$$

where $v : D \to \mathbb{R}^H$ is a hidden state, $\sigma$ is a standard activation function (e.g., GELU), $W \in \mathbb{R}^{H \times H}$ is a weight matrix, $b \in \mathbb{R}^H$ is a bias vector and $\kappa : D \to \mathbb{R}^{H \times H}$ is a learnable integral-kernel.

To give a unified discussion, we will say that $\Psi$ is a **convolutional NO**, if it is of the form $\Psi(u) = \mathcal{Q} \circ \mathcal{L}_L \circ \cdots \circ \mathcal{L}_1 \circ \mathcal{R}(u)$, with hidden layers $\mathcal{L}_\ell$ as in equation 3, and a choice of parametrized kernel $\kappa_\ell(x) = \kappa_\ell(x; \theta)$. In addition, we have a lifting layer $\mathcal{R}(u)(x) := R(u(x), x)$ and projection layer $\mathcal{Q}(v(x)) = Q(v(x))$ by composition with shallow neural networks $R, Q$. We next discuss three architectures exemplifying this approach, highlighting their commonality and giving a unified description. This results in a sharp *criterion for universality* for any such convolutional NO architecture.

**Fourier neural operator (FNO).** The FNO parametrizes the convolution kernel by a truncated Fourier series, $\kappa(x) = \sum_{|k|_\infty \le K} \hat{\kappa}_k e^{ikx}$, with cut-off parameter $K$, and $|k|_\infty = \max_{j=1,\dots,d} |k_j|$. The Fourier coefficients $\hat{\kappa}_k \in \mathbb{C}^{H \times H}$ are complex-valued matrices. The parametrization of the integral kernel $\kappa(x)$ of FNO in $d$ dimensions entails a considerable memory footprint, requiring $O(LH^2K^d)$ parameters. This can render FNO prohibitive in high-dimensional applications.

**Factorized FNO (F-FNO).** To lessen the computational demands of FNO, so-called "factorized" architectures have been introduced. Here, convolutions are taken along one dimension at a time. Mathematically, this corresponds to choosing kernels of the form $\kappa(x) = \kappa_s(x_j) \prod_{k \ne j} \delta(x_k)$, where $\kappa_s(x_j)$ is a 1d FNO kernel and $\delta(x_k)$ the Dirac delta function. Thus, integration is effectively only performed with respect to $x_j$. The resulting F-FNO only requires $O(LH^2Kd)$ parameters Tran et al. (2023).

**Spatial SSM.** We next consider the spatial convolution layers of the proposed SS-NO architecture. In this case, solution of the ODE system equation 1 leads to an explicit formula for the corresponding kernel. For a 1d spatial domain, this results in a kernel $\kappa(x) = \kappa_+(x) + \kappa_-(x)$, where $\kappa_\pm(x)$ correspond to the backward and forward scans, respectively, and $\kappa_\pm(x)$ is of the form

$$\kappa_\pm(x) = \mathbb{1}_{\mathbb{R}_\pm}(x) \sum_{k=1}^K e^{r_k|x|} e^{i\omega_k x} C_k B_k^T, \ r_k = \text{Real}(\lambda_k), \ \omega_k = \text{Imag}(\lambda_k), \ B_k, C_k \in \mathbb{R}^H. \tag{4}$$

Here $\mathbb{1}_{\mathbb{R}_\pm}(x)$ is the indicator function of $\mathbb{R}_\pm = \{\pm x \ge 0\}$. The main difference with FNO lies in the parametrization of the convolutional kernel $\kappa$, which now has tunable frequency parameters $\omega_k \in \mathbb{R}$, and tunable damping parameters $r_k < 0$. The parameter count of the resulting factorized spatial SSM architecture in $d$-dimensions requires at most $O(LH(H + K)d)$ parameters.

A comparison of the model parameter count entailed by the above choices is summarized below:

| Model | FNO | F-FNO | spatial-SSM |
|---|---|---|---|
| # Parameter (scaling) | $LH^2K^d$ | $LH^2Kd$ | $L(H^2 + HK)d$ |

### 4.1 A Sharp Criterion for Universality of Convolutional NOs.

As highlighted above, FNO, factorized FNO and SSMs all share a common structure, distinguished by the chosen kernel parametrization. *What can be said about the expressivity of such architectures?* Our first goal is to derive a sharp, general condition for the universality of such convolutional NO architectures.

A minimal requirement for the universality of neural network architectures is that the value of each output pixel must be informed by the values of *all* input pixels, i.e., the model needs to have a "full field of view". For convolutional NO architectures, this leads to the following definition (cf. discussion in App. B.2):

**Definition.** *A convolutional NO architecture with layer kernels $\kappa_1, \ldots, \kappa_L$ has a **full field of view**, if the iterated kernel $\bar{\kappa} := \kappa_L * \kappa_{L-1} * \cdots * \kappa_1$ is non-vanishing, i.e. $\bar{\kappa}(x - y) \neq 0$ for all $x, y \in D$.*

The next result shows that this "minimal" condition is actually already sufficient for the universality of a convolutional architecture, requiring no additional assumptions on the convolution operators:

**Theorem 4.1.** *A (factorized) convolutional NO architecture is universal if it has a full field of view.*

We refer to Appendix B for the precise statement and proof. To the best of our knowledge, the criterion identified above is both the simplest and most widely applicable result for convolutional NO architectures; it implies universality of (vanilla) FNO, factorized FNOs and combinations of (F-)FNO and SSMs, SS-NO and even gives a sharp criterion for the universality of *localized* convolutional NOs (similar to the CNO Raonic et al. (2023)), for which $\kappa_\ell$ has localized support. A theoretical basis for both factorized and localized architectures had been missing from the literature. The above result closes this gap. Appendix C.1 contrasts empirical results with a unidirectional SSM that violates the criterion.

### 4.2 Adaptivity and Enhanced Model Expressivity of SS-NO

Based on the formulation in equation 4, the factorized spatial SSM structurally subsumes the FNO (in 1D) and F-FNO architectures. We provide a rigorous proof of this capability in Appendix I. Fundamentally, the SSM parameterizes a continuous 1D convolutional kernel of the form

$$\kappa(x) = \sum_{k=1}^{K} c_k e^{-\rho_k |x|} e^{i\omega_k x}, \quad \rho_k \geq 0, \quad \omega_k \in \mathbb{R}. \tag{5}$$

This formulation generalizes the standard Fourier basis, allowing the model to learn adaptive frequencies and damping rates beyond the fixed harmonics of the DFT. However, since the damping and frequency are *tunable parameters* within SS-NO, this adds further adaptivity: (1) choice of $\rho_k > 0$ allows effective *kernel localization*; the model can optimize the support of its convolutional kernel, interpolating between the global kernels of FNO ($\rho_k \approx 0$) and very localized CNN-like kernels ($\rho_k \gg 1$). (2) additional adaptivity comes from *adaptive mode-filtering*; the model can optimize the frequencies $\omega_1, \ldots, \omega_K$, most relevant for nonlocal processing. Thus, in theory, this added adaptivity implies enhanced model expressivity of SS-NO over F-FNO.

## 5 Experiment Setup

### 5.1 Setup and Dataset Description

We evaluate our models on a suite of one-dimensional (1D) and two-dimensional (2D) partial differential equation (PDE) benchmarks commonly used in operator learning. These datasets span a variety of dynamical regimes—from chaotic behavior to turbulent flows—and are designed to assess the models' ability to capture long-range temporal dependencies. For 1D problems, we use datasets based on the Burgers' equation (nonlinear convection–diffusion) and the Kuramoto–Sivashinsky equation (chaotic dynamics), following the same data sources as Buitrago et al. (2025) and generating low-resolution versions through uniform spatial downsampling.

For 2D problems, we evaluate several widely used benchmarks, including the `TorusLi`, `TorusVis`, and `TorusVisForce` datasets (Li et al., 2021; Tran et al., 2023), as well as compressible Euler benchmarks such as the Richtmeyer–Meshkov instability (CE-RM) and the Rayleigh–Taylor instability with gravitational forcing (GCE-RT) (Herde et al., 2024). The `TorusVis` and `TorusVisForce` datasets incorporate variable, time-dependent forcing under randomly sampled viscosities in the range $\nu \in [10^{-5}, 10^{-4}]$. Together, these benchmarks test the models' ability to capture turbulence, long-range interactions, shocks, chaotic dynamics, and interface instabilities.

For evaluation, we compare SS-NO against a suite of popular baseline models including U-Net (Gupta & Brandstetter, 2023), GKT (Cao, 2021), Factformer (Li et al., 2023a), and FFNO (Tran et al., 2023; Buitrago et al., 2025) for 1D benchmarks, and additionally FNO with a two-dimensional kernel (2D FNO) (Li et al.,

2021) for 2D problems. Following the MemNO framework (Buitrago et al., 2025), we augment all baselines and SS-NO with a temporal S4 module using a memory window of $K = 4$ for fair comparison. The only exception is GKT, for which we use a multi-input variant with $K = 4$ where the temporal dimension is mixed with features, as we found the model struggled to converge with an additional temporal S4 module.

**Data Preprocessing and Training Setup.** We follow standard practices from prior work (Tran et al., 2023), normalizing all data to the range $[0, 1]$ and adding Gaussian noise with variance $10^{-3}$ during training for stability. Models are trained to minimize the step-wise normalized relative $\ell_2$ loss.

Unless otherwise specified, all models use four blocks with consistent hidden dimensions. Full details on baseline models, data generation, preprocessing, and training are provided in Appendices E and F.

### 5.2 Training Objective.

We train the model by minimizing the empirical risk over the dataset of trajectories. Given a loss function $\ell : L^2(\Omega; \mathbb{R}^V) \times L^2(\Omega; \mathbb{R}^V) \to \mathbb{R}$, we solve:

$$\theta^* = \arg\min_{\theta} \frac{1}{N} \sum_{i=1}^{N} \frac{1}{N_{\text{out}}} \sum_{t \in \mathcal{T}_{\text{out}}} \ell\left(u^{(i)}(t, x), \ \Psi_{\theta, t}\left(u^{(i)}(t', x)|_{t' \in \mathcal{T}_{\text{in}}}\right)\right), \tag{6}$$

where $\Psi_{\theta, t}(\cdot)$ denotes the prediction at time $t \in \mathcal{T}_{\text{out}}$. As our loss $\ell$, we employ the relative $\mathcal{L}^2$-error, discussed below. The formulation in equation 6 enables the model to learn the entire future evolution of the system dynamics from a finite observed window, rather than stepwise or autoregressive forecasting.

### 5.3 Evaluation Metric.

Performance is reported using the relative $\ell_2$ error:

$$Relative \ \ell_2(u(t, x), \hat{u}(t, x)) = \frac{\|u(t, x) - \hat{u}(t, x)\|_2}{\|u(t, x)\|_2}, \tag{7}$$

where $\| \cdot \|_2$ denotes the squared norm over all spatial locations and time steps in the output horizon.

All results reported in the main text represent the mean performance, measured by the **best validation loss**, across **five independent runs** with different random seeds. To contextualize performance relative to model complexity, we also report the **number of parameters** for each model, defined as the total count of learnable parameters. We note that for all 1D experiments, the standard deviation is less than $\pm 5\%$ of the reported mean at resolutions 128 and 64, and less than $\pm 7\%$ at resolution 32, while for all 2D experiments, the standard deviation is below $\pm 0.3\%$ in relative $\ell_2$ error. The exceptions to this are GKT and Factformer (2D), which exhibit slightly higher variability.

## 6 Results

### 6.1 1D Burgers' Equation

We evaluate model performance on the canonical 1D Burgers' equation with $\nu = 0.001$ across temporal resolutions of 128, 64, and 32. As shown in Figure 3, SS-NO achieves competitive accuracy at all resolutions while demonstrating exceptional parameter efficiency.

Notably, SS-NO delivers superior performance across all tested regimes. Unlike Fourier-based models whose complexity often scales with input resolution and the number of spectral modes, SS-NO maintains a consistent architectural footprint independent of the input grid size. This structural advantage is particularly valuable for applications requiring seamless deployment across multiple spatial or temporal scales.

The Burgers' equation exhibits smooth dynamics dominated by diffusion and mild nonlinearity, and SS-NO's consistent performance across resolutions highlights its robustness in handling such well-behaved systems. The model's ability to maintain accuracy while drastically reducing parameter count represents a significant advancement in efficient operator learning.

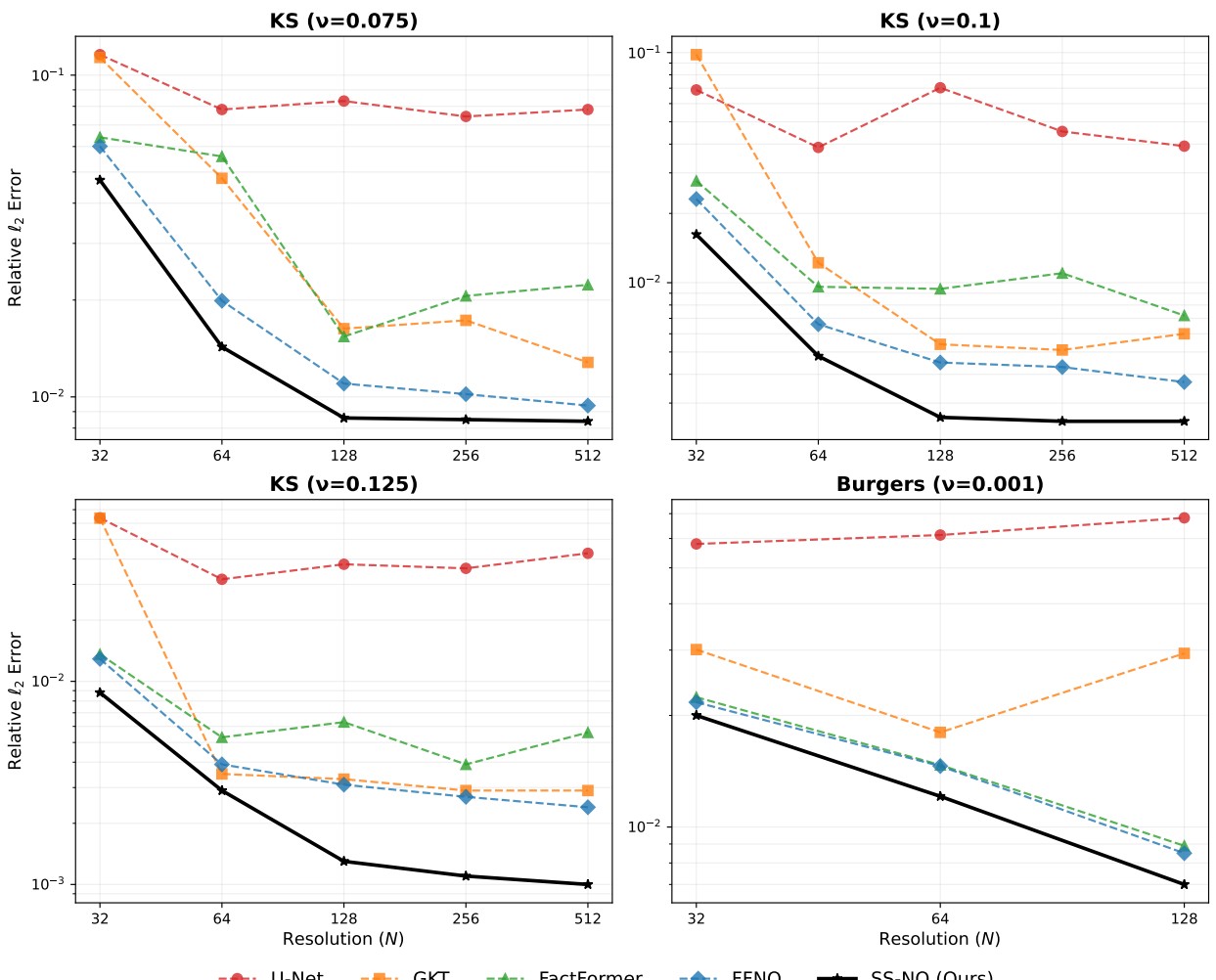

Figure 3: **Resolution Dependence Analysis.** Relative $\ell_2$ error of SS-NO and baseline models on 1D benchmarks across varying spatial resolutions ($N \in \{32, \ldots, 512\}$). **Left three panels:** Kuramoto-Sivashinsky (KS) equation with increasing viscosity coefficients $\nu \in \{0.075, 0.1, 0.125\}$. **Right panel:** Burgers' equation ($\nu = 0.001$), limited to $N = 128$ due to dataset constraints. While U-Net performance stagnates at higher resolutions ($N \geq 128$), operator learning methods generally improve. Notably, **SS-NO (Ours)** achieves superior accuracy early at $N = 128$ and maintains the lowest error floor at $N = 512$, demonstrating robust resolution efficiency.

## 6.2 1D Kuramoto–Sivashinsky Equation

We next evaluate models on the 1D Kuramoto–Sivashinsky (KS) equation across three viscosities ($\nu = 0.075, 0.1, 0.125$). The KS equation presents a challenging benchmark characterized by increasingly chaotic dynamics and multiscale spatiotemporal features as viscosity decreases.

**Performance Analysis.** As shown in Figure 3, SS-NO demonstrates consistent performance improvements, achieving the best results among evaluated methods across all viscosity regimes. At the standard resolution of $N = 128$, SS-NO outperforms the second-best model (FFNO) by 22% at $\nu = 0.075$, 42% at $\nu = 0.1$, and 58% at $\nu = 0.125$. These improvements are particularly pronounced in the low-viscosity regime ($\nu = 0.075$), where chaotic behavior dominates and traditional methods often struggle to track long-term dependencies.

**Resolution Dependence and Spectral Efficiency.** To analyze the asymptotic behavior of the architectures, we extended our evaluation to higher spatial resolutions of $N = 256$ and $N = 512$. In general, we observe diminishing returns in performance improvement for most architectures beyond $N = 128$, suggesting that the macroscopic dynamics of the system are largely resolved at this scale. However, a distinct trend emerges when comparing spectral and state-space approaches:

1. **Explicit Spectral Scaling (FFNO):** The FFNO demonstrates a slight but continuous reduction in error at higher resolutions. We attribute this to the explicit nature of the Fast Fourier Transform; as the grid resolution increases, the Nyquist frequency rises, allowing the FFNO to capture high-frequency details that were previously truncated.

2. **Implicit Adaptive Learning (SS-NO):** In contrast, SS-NO achieves its optimal performance floor earlier, effectively converging at $N = 128$, and maintains this superior accuracy at $N = 512$ even with a fixed state dimension. This stability supports our hypothesis regarding the *adaptive frequency learning* capabilities of State-Space Models. Unlike FFNO, which requires finer spatial discretization to resolve high-frequency features via FFT, SS-NO's recurrent mechanism appears capable of extracting and encoding these critical dynamics efficiently within its latent state space.

This efficiency is further evidenced at coarse resolutions. At $N = 32$, SS-NO maintains a 21% improvement over FFNO at $\nu = 0.075$ and 30% at $\nu = 0.1$. These results collectively demonstrate that SS-NO achieves an optimal balance of accuracy and robustness—excelling in chaotic regimes while maintaining parameter efficiency across varying resolutions.

## 6.3 Ablation Study

We conduct a comprehensive ablation study to understand the individual contributions of model capacity, learnable frequencies, and explicit damping mechanisms. Four distinct configurations are evaluated: **All**, where both frequencies and damping are learnable; **Damping only**, where frequencies remain fixed while damping is learnable; **Frequency only**, where frequencies are learnable but damping is explicitly set to zero (absent); and **Fixed**, where neither frequencies nor damping are learnable.

Experiments are conducted on three variants of the KS equation with viscosities $\nu \in \{0.075, 0.1, 0.125\}$, all at resolution 128. Performance is quantified using relative $\ell_2$ error. Results are presented in Table 1 to isolate the effects of architectural components across different capacity regimes.

Table 1: Relative $\ell_2$ error on KS benchmarks at varying state sizes and training settings.

| State Size | Setting | # Parameters | Relative $\ell_2$ Error | | |
|---|---|---|---|---|---|
| | | | KS | | |
| | | | $\nu = 0.075$ | $\nu = 0.1$ | $\nu = 0.125$ |
| 64 | All | 203,713 | 0.0086 | 0.0026 | 0.0013 |
| | Damping only | 187,329 | 0.0086 (+0.0000) | 0.0027 (+0.0001) | 0.0013 (+0.0000) |
| | Frequency only | 187,329 | 0.0110 (+0.0024) | 0.0030 (+0.0003) | 0.0014 (+0.0001) |
| | Fixed | 170,945 | 0.0116 (+0.0030) | 0.0033 (+0.0006) | 0.0016 (+0.0003) |
| 32 | All | 154,561 | 0.0090 | 0.0027 | 0.0012 |
| | Damping only | 146,369 | 0.0094 (+0.0004) | 0.0028 (+0.0001) | 0.0014 (+0.0002) |
| | Frequency only | 146,369 | 0.0137 (+0.0047) | 0.0038 (+0.0011) | 0.0018 (+0.0006) |
| | Fixed | 138,177 | 0.0158 (+0.0068) | 0.0042 (+0.0015) | 0.0020 (+0.0008) |
| 16 | All | 129,985 | 0.0115 | 0.0038 | 0.0017 |
| | Damping only | 125,889 | 0.0131 (+0.0016) | 0.0044 (+0.0006) | 0.0019 (+0.0002) |
| | Frequency only | 125,889 | 0.0198 (+0.0083) | 0.0069 (+0.0031) | 0.0031 (+0.0014) |
| | Fixed | 121,793 | 0.0227 (+0.0112) | 0.0068 (+0.0030) | 0.0031 (+0.0014) |

### 6.3.1 Observations

**Explicit damping enables remarkable parameter efficiency.** The most striking finding is that proper damping mechanisms allow models to maintain strong performance even under extreme capacity constraints. The 16-state damping-only model achieves performance competitive with much larger models, demonstrating that damping serves as a powerful regularization and stabilization mechanism that dramatically improves parameter efficiency. This efficiency is particularly evident in challenging regimes ($\nu = 0.075$), where the 16-state damping-only model approaches the performance of fixed-configuration models with 2-4× more states, despite its significantly reduced capacity.

**Model capacity dramatically affects damping requirements.** Higher-capacity models can implicitly compensate for architectural limitations through additional spectral modes, while lower-capacity models rely critically on explicit damping for stability. The performance degradation in 16-state models without damping versus 64-state models confirms that damping becomes increasingly essential as model capacity decreases. This progressive dependency relationship underscores damping's role as a crucial stabilization mechanism in capacity-constrained settings.

**Damping significance escalates with problem difficulty.** The performance gap between models with and without damping grows substantially with increasing chaos (decreasing viscosity). For $\nu = 0.075$, proper damping provides significant absolute improvement, while for $\nu = 0.125$, the benefit reduces substantially. This differential effect confirms that chaotic dynamics benefit substantially from explicit damping mechanisms, while smoother regimes can tolerate their absence.

**Learnable frequencies provide consistent spectral adaptation benefits.** Across all configurations, the frequency-only setting consistently outperforms the fixed configuration, demonstrating the value of data-driven spectral basis adaptation. This benefit is particularly pronounced in constrained settings and complex regimes like KS $\nu = 0.075$, where learnable frequencies allow models to dynamically allocate their limited spectral resources to the most relevant temporal modes. Rather than being constrained to a fixed Fourier basis, models can adapt their frequency representations to focus on the specific oscillatory patterns most critical for the target dynamics, leading to more efficient temporal representation learning. While these gains are generally more modest than those provided by explicit damping—particularly in challenging regimes—they confirm that learnable frequencies provide meaningful improvements in spectral efficiency.

**Results are not driven by overparameterization.** The consistent performance pattern across capacity levels confirms that our findings are not artifacts of excessive model size. Even the smallest models (16 states) achieve competitive performance when equipped with appropriate inductive biases, demonstrating that architectural design rather than parameter count drives performance improvements.

### 6.3.2 Damping Analysis: Connecting Ablation Results to Learned Behavior

The ablation results naturally raise the question: how do these performance differences manifest in the actual learned damping behavior? To answer this, we conduct a detailed analysis of the learned damping distributions across different configurations, aggregating results from five independently trained models with different random seeds for each case to ensure statistical robustness.

**Architectural constraints drive stronger damping.** Quantitative analysis reveals that damping-only models learn 24% **stronger mean damping** than full models (0.726 vs. 0.586, $p = 5.2 \times 10^{-118}$), representing a 163% **larger relative increase** from initialization (45.3% vs. 17.2% increase from 0.5), as shown in Figure 4. This compensatory strengthening explains the ablation results: when frequency adaptation is disabled, models intensify their damping mechanisms to maintain stability, directly corresponding to the observed patterns in Table 1.

**Problem difficulty modulates damping strength.** Models trained on more chaotic dynamics ($\nu = 0.075$) learn 14% **stronger damping** than those on smoother regimes ($\nu = 0.125$, 0.586 vs. 0.514, $p = 5.8 \times 10^{-71}$), with a 537% **larger relative increase** from initialization (17.2% vs. 2.7% increase from 0.5), as visualized in Figure 5. This dramatic differential learning directly mirrors the ablation findings where damping provides exponentially greater benefits in challenging regimes.

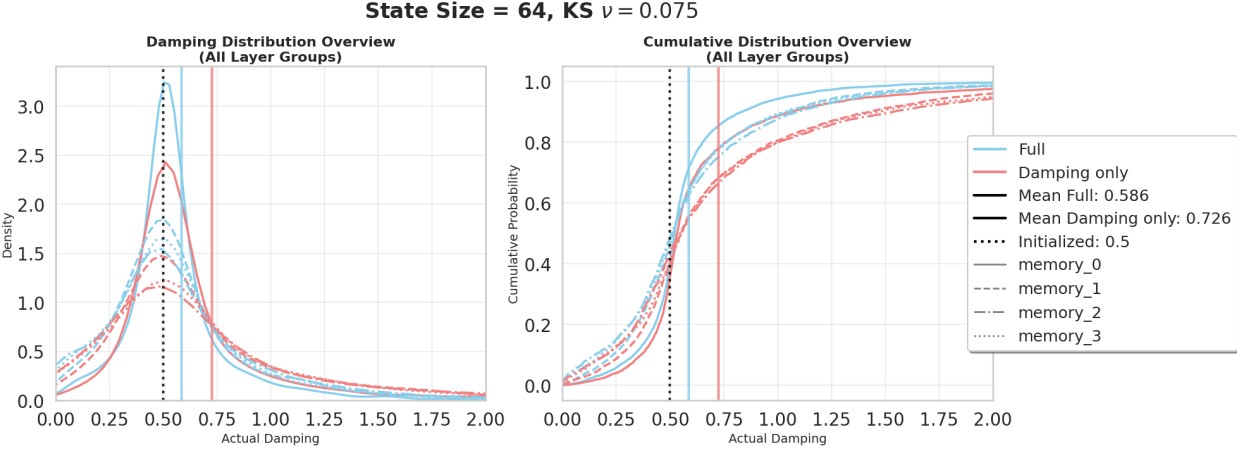

Figure 4: Damping coefficient distributions for full vs. damping-only models at $\nu = 0.075$ with 64 states.

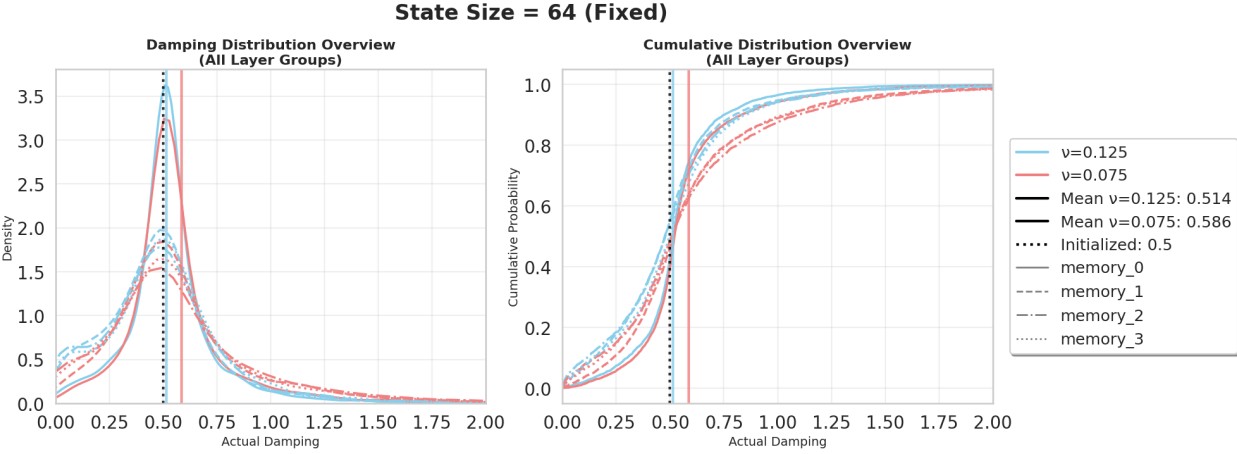

Figure 5: Damping coefficient distributions for $\nu = 0.125$ vs. $\nu = 0.075$ models with 64 states.

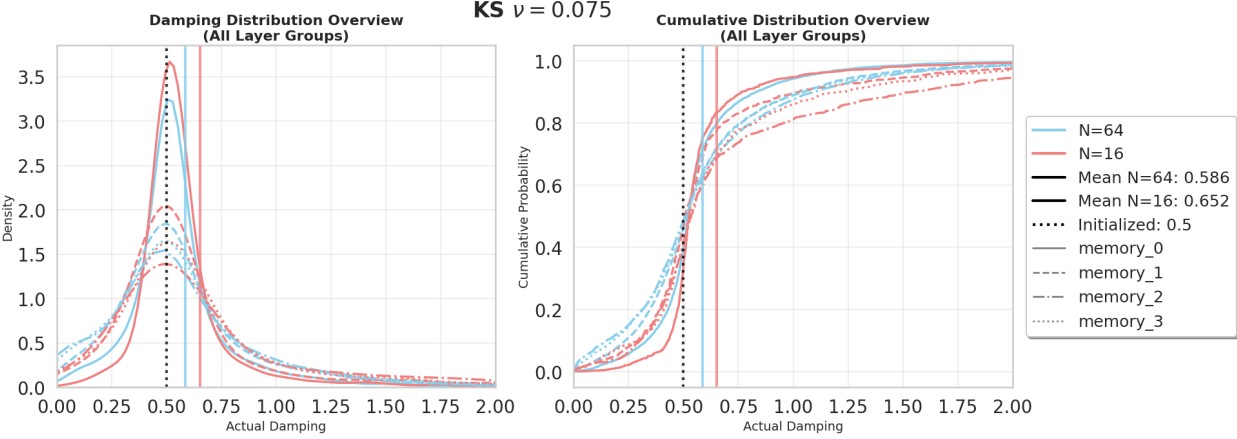

Figure 6: Damping coefficient distributions at $\nu = 0.075$ for different state sizes.

**Capacity constraints amplify damping requirements.** Lower-capacity models (16 states) learn 11% **stronger damping** than higher-capacity counterparts (64 states, 0.652 vs. 0.586, $p = 2.2 \times 10^{-15}$), with a

77% **larger relative increase** from initialization (30.4% vs. 17.2% increase from 0.5), as demonstrated in Figure 6. This intensified damping in capacity-constrained models explains their ability to maintain stability despite reduced parameter counts, directly connecting to the efficiency results in our ablation study.

The consistent pattern across all three analyses—supported by overwhelming statistical significance ($p < 10^{-15}$ in all cases)—demonstrates that **damping serves as a crucial stabilization mechanism that scales intelligently with architectural and problem constraints**. Models automatically learn to strengthen damping in response to constraints, providing a mechanistic explanation for the performance patterns observed in our ablation study. This adaptive damping behavior represents a key innovation that enables both parameter efficiency and robustness across diverse dynamical regimes.

Table 2: Relative $\ell_2$ error on 2D Navier–Stokes datasets at $64 \times 64$ resolution.

| Architecture | # Parameters | Relative $\ell_2$ Error | | | | |
|---|---|---|---|---|---|---|
| | | TorusLi | TorusVis | TorusVisForce | CE-RM | GCE-RT |
| U-Net | $7,783,777$ | 0.0611 | 0.0474 | 0.0558 | 0.0644 | 0.0300 |
| Factformer (2D) | $1,006,433$ | 0.0628 | 0.0337 | 0.0395 | 0.0642 | 0.0315 |
| GKT | $8,418,049$ | 0.0619 | 0.0404 | 0.0742 | 0.0691 | 0.0172 |
| FNO2D | $67,197,700$ | 0.0452 | 0.0466 | 0.0444 | 0.0717 | 0.0155 |
| FFNO | $2,192,897$ | 0.0409 | 0.0231 | 0.0326 | 0.0688 | 0.0196 |
| SS-NO (ours) | $369,665$ | **0.0345** | **0.0218** | **0.0263** | **0.0583** | **0.0138** |

### 6.4 2D Navier–Stokes Equations

**Navier–Stokes with Fixed Viscosity.** We evaluate our model on the `TorusLi` dataset, which contains vorticity fields generated by solving the 2D incompressible Navier–Stokes equation on the unit torus with a fixed forcing term and low viscosity $\nu = 10^{-5}$. This regime is particularly challenging due to reduced diffusion, leading to highly nonlinear and chaotic dynamics.

Table 2 reports the relative $\ell_2$ errors of various baselines. Among them, FNO2D achieves an error of 0.0452. In contrast, SS-NO significantly outperforms this baseline, reducing the error to 0.0345—representing a 23.7% improvement in predictive accuracy.

This performance gap is notable given the differences in scaling. FNO2D leverages a full 2D Fourier transform to model frequency interactions jointly across both spatial dimensions, with cost scaling as $\mathcal{O}(N^D)$, where $D$ is the number of dimensions and $N$ is the resolution per dimension. In contrast, SS-NO applies separate 1D operators along each axis in a factorized manner, achieving more favorable $\mathcal{O}(N \cdot D)$ complexity while retaining or improving accuracy. These results demonstrate the advantage of temporal-state modeling even in time-invariant contexts, highlighting the role of state-space modeling for capturing fluid dynamics in low-viscosity regimes. Among several factorized connection variants we tested, our configuration delivered the best performance (see Appendix D).

**2D Navier-Stokes: Problems with Varying Viscosities and Forces.** We also evaluate our model on the `TorusVis` and `TorusVisForce` datasets from the FFNO benchmark (Tran et al., 2023), where both viscosity and external forcing vary across trajectories. In `TorusVis`, viscosity is sampled uniformly between $10^{-5}$ and $10^{-4}$ and the forcing function is time-independent. `TorusVisForce` introduces a time-varying forcing function with phase shift $\delta = 0.2$.

Table 2 summarizes the results. SS-NO achieves the lowest errors across both datasets, consistently outperforming all baselines. This demonstrates that SS-NO effectively captures spatiotemporal dynamics across mixed-viscosity regimes, where larger viscosity values in the data are significantly easier to predict, while still handling the challenging low-viscosity cases.

**Compressible Euler Benchmarks (CE-RM and GCE-RT).** We also evaluate SS-NO on compressible Euler benchmarks: the Richtmyer–Meshkov instability (CE-RM) and the Rayleigh–Taylor instability with

gravitational forcing (GCE-RT). These problems involve complex interface dynamics and shocks, making them particularly challenging for predictive modeling.

Table 2 shows that CE-RM has relatively high errors across all models, reflecting its intrinsic difficulty. Interestingly, on GCE-RT, GKT and FNO2D perform better compared to other factorized baselines, suggesting that explicitly modeling full 2D spatial interactions can be beneficial for this problem.

Despite being factorized, SS-NO achieves the lowest relative $\ell_2$ errors on both CE-RM (0.0583) and GCE-RT (0.0138), demonstrating that it can efficiently capture the dominant spatiotemporal dynamics even in problems with shocks and complex interface interactions. These results highlight that SS-NO combines the efficiency of factorized operations with strong modeling capacity, making it competitive with full 2D approaches in capturing challenging fluid behavior.

## 7 Conclusion and Future Work

We have presented the State Space Neural Operator (SS-NO), a compact neural operator designed to efficiently learn solution operators for time-dependent partial differential equations. Grounded in a theoretical universality result for convolutional operator learning with full field of view, SS-NO generalizes and improves upon prior factorizations such as FNO by integrating adaptive mechanisms—including damping for receptive field localization and learnable frequency modulation for data-driven mode selection—while maintaining temporal causality. Our spatial-only variant exactly recovers F-FNO, yet SS-NO dramatically reduces parameter count by exploiting a linear $\mathcal{O}(N \cdot D)$ scaling compared to the $\mathcal{O}(N^D)$ complexity of high-dimensional convolutions.

Empirically, SS-NO achieves competitive performance across a suite of 1D and 2D benchmarks, including the Burgers', Kuramoto–Sivashinsky, Navier–Stokes, and Euler equations, demonstrating strong accuracy with significantly fewer parameters. In addition, we proposed a dimensionally factorized variant of SS-NO, which scales favorably in higher dimensions and achieves competitive results on challenging 2D fluid dynamics datasets. These findings highlight that state space modeling is not only compatible with operator learning but also offers unique advantages in long-term memory, adaptivity, and efficiency.

While our formulation provides clear benefits for structured spatiotemporal domains, one limitation is the lack of a systematic connection between damping-based receptive field localization and irregular geometries. Developing principled extensions of SS-NO to unstructured meshes and complex domains remains an important direction for future work. Beyond this, we aim to apply SS-NO to higher-dimensional and multi-physics systems, including astrophysical, cosmological, and space plasma dynamics, where efficient and generalizable PDE operators are critical.

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

## A   Appendix

## B   Universality of Convolutional NOs

We here provide a rigorous definition of convolutional NOs having a "full field of view", and provide the precise statement of our sharp universality condition.

### B.1   Full Field of View

We describe the required full field of view property in greater detail, which we informally stated as a definition before Theorem 4.1 in the main text.

The following assumptions were implicit in that definition: The convolutional NO architecture is of the form

$$\Psi(u) = \mathcal{Q} \circ \mathcal{L}_L \circ \cdots \circ \mathcal{L}_1 \circ \mathcal{R}(u),$$

where the lifting layer $\mathcal{R}(u)(x) := R(u(x), x)$ and projection layer $\mathcal{Q}(v(x)) = Q(v(x))$ are given by composition with shallow neural networks $R, Q$ of width $H$ (both applied pointwise to the respective input functions). The hidden layers $\mathcal{L}_\ell$ are given by

$$\mathcal{L}_\ell(v)(x) = \sigma\left(W_\ell v(x) + b_\ell + \int_D \kappa_\ell(x - y; \theta_\ell)v(y)\, dy\right),$$

with channel width $H$. We will assume that $H$ can be chosen as large as required. For the integral kernel $\kappa_\ell$, we assume that, for any choice of $H$ and matrix $A_\ell \in \mathbb{R}^{H \times H}$ acting on the channel dimension, there exists a setting of the parameters $\theta_\ell^*$, such that $\kappa_\ell(x; \theta_\ell^*) \in \mathbb{R}^{H \times H}$ is a scalar multiple of $A$, i.e. the kernel $\kappa_\ell(\,\cdot\,; \theta_\ell^*)$ is the form

$$\kappa_\ell(x; \theta_\ell^*) \in \mathbb{R}^{H \times H} = g_\ell(x)\, A_\ell, \quad g_\ell : D \to \mathbb{R} \text{ some scalar function.} \tag{8}$$

To our knowledge, all proposed architectures, including FNO, factorized FNO, SSM, as well as (UNet-style) convolutional NOs with localized kernels have this property.

The precise definition of the full field of view property is then the following:

**Definition** (rigorous)**.** *An architecture is said to have a full field of view, if for any channel width $H$ and $A_\ell \in \mathbb{R}^{H \times H}$ with product $A_L \cdots A_1 \neq 0$, we can find parameters such that equation 8 holds, and the iterated kernel*

$$\bar{\kappa}(x) = (\kappa_L * \cdots * \kappa_1)(x) \equiv (g_L * \cdots * g_1)(x)\, A_L \cdots A_1, \tag{9}$$

*has full support, in the sense that the support[1] of the function $(x, y) \mapsto \bar{\kappa}(x - y)$ is $D \times D$.*

**Remark 1.** In particular, $\bar{\kappa}$ satisfies the full field of view property, if $\bar{\kappa}$ is nowhere-vanishing, i.e. for any $x, y \in D$, we have $\bar{\kappa}(x - y) \neq 0$.

### B.2   Intuitive Explanation of the Full Field of View Property

Before stating our universality theorem, based on the "full field of view" property, we discuss its intuitive meaning in this section.

A general (non-linear) operator $\mathcal{G} : u \mapsto \mathcal{G}(u)$ takes an input function $u(x)$ and maps it to another function $v(x) = \mathcal{G}(u)(x)$. In general, such operators are *non-local*, meaning that the value of the output function $v(x)$ at a point $x$ depends on the values of the input function $u(y)$, for $y \in D$, across the entire domain $D$.

A simple example of an operator with non-local dependency is an operator with a general integral kernel:

$$\mathcal{G}(u)(x) := \int_D k(x, y)u(y)\, dy.$$

---

[1]We recall that the support of a general function $f(z)$ is the closure of the set $\{z \mid f(z) \neq 0\}$.

To determine the value of the output function at $x$ for such $\mathcal{G}$, it is clear that we generally require knowledge of $u(y)$ at all points $y \in D$. While this illustrative example defines a linear operator $\mathcal{G}$, more general forms of non-locality are present in most operators of interest (such as solution operators of non-linear PDEs), which generally are both non-local and non-linear.

Thus, it is intuitively clear that a universal operator-learning architecture must provide a mechanism that enables information to flow from the input function $u(y)$ at any point $y \in D$ to the output function $v(x)$ at any $x \in D$.

For the convolutional architectures discussed in this work, the only mechanism to communicate information non-locally during the forward pass is via the convolutional integral kernels $v(x) \mapsto \int_D \kappa_\ell(x-y)v(y)\,dy$. The iterated kernel equation 9 is a mathematical way to capture the flow of information across all layers: Indeed, if $\bar{\kappa}(x-y) \neq 0$, this means that (at least some) information about the input function at $y$ can flow to the output function at $x \in D$. However, a priori, it is unclear whether this specific mechanism to transfer information non-locally is sufficiently informative. Indeed, one might assume that the $x$-dependency of the kernels $\kappa_\ell(x) = g_\ell(x)A_\ell$, i.e. the functions $g_\ell(x)$ themselves, need to be very specifically tuned to ensure all relevant information on the input side to correctly reach the output side.

The main theoretical insight of the present work is to show that no specific tuning of $g_\ell(x)$ is required for universality. In fact, *any* functions $g_1, \ldots, g_L$ will do, as long as they ensure a full field of view and as long as we can tune the matrices $A_1, \ldots, A_L \in \mathbb{R}^{H \times H}$ (and allow arbitrarily large hidden channel dimension $H$). Given this discussion, we view the "full field of view" property as a "minimal" property to ensure the needed flow of information through our architecture. In Theorem B.1 below, we will rigorously prove that this "minimal" property is already sufficient.

### B.3 A Sufficient Condition for Universality

We can now state and prove our universality result for general convolutional NOs.

**Theorem B.1.** *Let $\mathcal{G} : C(D;\mathbb{R}^{d_i}) \to C(D;\mathbb{R}^{d_o})$ be a continuous operator, with either $D \subset \mathbb{R}^d$ compact or $D = \mathbb{T}^d$ the periodic torus. Let $\mathcal{K} \subset C(D;\mathbb{R}^{d_i})$ be a compact set of input functions. Let $\Psi(u)$ be a convolutional NO architecture with a full field of view. Then for any $\epsilon > 0$, there exists a channel width $H$, and a setting of the weights of $\Psi$, such that*

$$\sup_{u \in \mathcal{K}} \sup_{x \in D} |\mathcal{G}(u)(x) - \Psi(u)(x)| \leq \epsilon.$$

#### B.3.1 Proof of sufficiency of "full field of view" property (Theorem B.1)

*Proof.* For simplicity, we will assume that $d_i = d_o = 1$. The proof easily extends to the more general case. The space of continuous operators $\mathcal{G} : \mathcal{K} \to C(D)$, $u(x) \mapsto \mathcal{G}(u)(x)$, is isometrically isomorphic to the space $C(\mathcal{K} \times D)$, by identifying $\mathcal{G}(u,x) := \mathcal{G}(u)(x)$. By assumption, both $\mathcal{K}$ and $D$ are compact sets.

Given the choice of convolutional NO architecture, we now fix $\kappa_1, \ldots, \kappa_L$ such that $\bar{\kappa}$ is nowhere vanishing. Let us now introduce the set $\mathbb{A}$ consisting of all operators that can be represented by a choice of channel width $H$ and a choice of tunable parameters:

$$\mathbb{A} := \left\{ \begin{array}{c} \Psi = \mathcal{Q} \circ \mathcal{L}_L \circ \cdots \circ \mathcal{L}_1 \circ \mathcal{R}, \ \Psi \text{ is a convolutional NO} \\ \text{with kernels } \kappa_1, \ldots, \kappa_L \text{ and channel width } H \end{array} \right\}.$$

Our goal is to show that $\mathbb{A} \subset C(\mathcal{K} \times D)$ is dense. We denote by $\overline{\mathbb{A}}$ the closure of $\mathbb{A}$ in $C(\mathcal{K} \times D)$ (set of limit points). To prove that $\mathbb{A} \subset C(\mathcal{K} \times D)$ is dense, we will use the following result, which follows from the Stone-Weierstrass theorem:

**Lemma B.2.** *A subset $\mathbb{A} \subset C(\mathcal{K} \times D)$ is dense, if*

- *The constant function $1 \in \mathbb{A}$,*

- *$\mathbb{A}$ separates points: For any $(u_1, x_1), (u_2, x_2) \in \mathcal{K} \times D$, there exists $\Psi \in \mathbb{A}$ such that $\Psi(u_1, x_1) \neq \Psi(u_2, x_2)$.*

- $\mathbb{A}$ *is an approximate vector subalgebra:*
  - $\mathbb{A}$ *is closed under addition and scalar multiplication (i.e. vector subspace),*
  - $\mathbb{A}$ *is approximately closed under multiplication: For any* $\Psi_1, \Psi_2$, *the product* $\Psi_1 \cdot \Psi_2 \in \overline{\mathbb{A}}$, *where*

  $$(\Psi_1 \cdot \Psi_2)(u, x) = \Psi_1(u, x)\Psi_2(u, x),$$

  *is the pointwise multiplication.*

Our goal is to show these properties for $\mathbb{A}$ to conclude that $\mathbb{A} \subset C(\mathcal{K} \times D)$ is dense.

$\mathbb{A}$ **contains constants.**   It is very easy to show that the constant function $1 \in \mathbb{A}$, by defining a convolutional NO that disregards the input and has constant output $= 1$.

$\mathbb{A}$ **is an approximate subalgebra.**   To show the other properties, we first note that $\mathbb{A}$ is closed under scalar multiplication and under addition, i.e.

$$\Psi_1, \Psi_2 \in \mathbb{A} \quad \Rightarrow \quad \lambda_1 \Psi_1 + \lambda_2 \Psi_2 \in \mathbb{A}, \quad \forall \lambda_1, \lambda_2 \in \mathbb{R}.$$

If $H_1$ and $H_2$ are the channel widths of $\Psi_1$ and $\Psi_2$, respectively, this conclusion follows by a simple parallelization of $\Psi_1$ and $\Psi_2$, and employing the last projection $\mathcal{Q}$-layer to sum (and scale) the parallelized results. Thus, $\mathbb{A}$ is a vector subspace of $C(\mathcal{K} \times D)$.

To show that $\mathbb{A}$ is approximately closed under multiplication, i.e. $\Psi_1, \Psi_2 \in \mathbb{A}$ implies that $\Psi_1 \cdot \Psi_2 \in \overline{\mathbb{A}}$, we fix arbitrary $\Psi_1, \Psi_2 \in \mathbb{A}$. We will denote by $\hat{\Psi}_j$ the NO $\Psi_j$ for $j = 1, 2$, but where the last $\mathcal{Q}$ projection-layer has been removed:

$$\Psi_1 \equiv \mathcal{Q}_1 \circ \hat{\Psi}_1, \qquad \Psi_2 \equiv \mathcal{Q}_2 \circ \hat{\Psi}_2.$$

Assuming wlog that the channel width $H_1 = H_2 = H$ (otherwise, we can pad the channel width by zeros), we note that these incomplete NOs $\hat{\Psi}_1$ and $\hat{\Psi}_2$ define continuous mappings $\mathcal{K} \times D \to \mathbb{R}^H$. Since $\mathcal{K} \times D$ is compact, it follows that also the images $\hat{\Psi}_j(\mathcal{K} \times D) \subset \mathbb{R}^H$ are compact (since compact sets are mapped to compact sets under continuous maps). In particular, there exists $B > 0$, such that

$$\hat{\Psi}_1(u, x), \hat{\Psi}_2(u, x) \in [-B, B]^H, \quad \forall\, u \in \mathcal{K}, x \in D.$$

Consider now $\mathcal{Q}_1$ and $\mathcal{Q}_2$, i.e. the last layers of $\Psi_1$ and $\Psi_2$, respectively. The following product mapping

$$[-B, B]^H \times [-B, B]^H \to \mathbb{R}, \quad (v_1, v_2) \to \mathcal{Q}_1(v_1) \cdot \mathcal{Q}_2(v_2),$$

is continuous. By the universality of shallow neural networks, for any $\epsilon > 0$, there thus exists a shallow net $\mathcal{Q} : \mathbb{R}^{2H} \to \mathbb{R}$, such that

$$\sup_{|v_1|_\infty, |v_2|_\infty \leq B} |\mathcal{Q}(v_1, v_2) - \mathcal{Q}_1(v_1) \cdot \mathcal{Q}_2(v_2)| \leq \epsilon.$$

Define now a new NO as $\Psi(u, x) := \mathcal{Q}([\hat{\Psi}_1(u, x), \hat{\Psi}_2(u, x)])$, i.e. $\mathcal{Q}$ applied to the parallelization of $\hat{\Psi}_1$ and $\hat{\Psi}_2$. Then

$$\sup_{\mathcal{K} \times D} |\Psi(u, x) - \Psi_1(u, x) \cdot \Psi_2(u, x)|$$
$$= \sup_{\mathcal{K} \times D} |\mathcal{Q}([\hat{\Psi}_1(u, x), \hat{\Psi}_2(u, x)]) - \mathcal{Q}_1(\hat{\Psi}_1(u, x)) \cdot \mathcal{Q}_2(\hat{\Psi}_2(u, x))|$$
$$\leq \sup_{|v_1|_\infty, |v_2|_\infty \leq B} |\mathcal{Q}([v_1, v_2]) - \mathcal{Q}_1(v_1) \cdot \mathcal{Q}_2(v_2)|$$
$$\leq \epsilon.$$

Since $\epsilon > 0$ was arbitrary, this shows that $\Psi_1 \cdot \Psi_2$ is a limit point of $\Psi \in \mathbb{A}$, thus $\Psi_1 \cdot \Psi_2 \in \overline{\mathbb{A}}$. We conclude from the above that $\mathbb{A}$ is an approximate vector subalgebra.

$\mathbb{A}$ **separates points.** To conclude our proof, it only remains to show that $\mathbb{A}$ separates points. Let $(u_1, x_1) \neq (u_2, x_2)$ be two distinct elements in $\mathcal{K} \times D$. There are two cases: Either $x_1 \neq x_2$, or $x_1 = x_2$ and $u_1 \neq u_2$.

**Case 1: $x_1 \neq x_2$.** We want to construct $\Psi \in \mathbb{A}$, such that $\Psi(u_1, x_1) \neq \Psi(u_2, x_2)$. This is easy, since we can always use the lifting layer $\mathcal{R}$ to eliminate the dependency of $\Psi$ on the $u$-variable. Once the weights acting on the $u$-variable have been set to zero, we then have that $\Psi(u, x) = \psi(x)$ is an ordinary multilayer perceptron. Since $x_1 \neq x_2$, we can easily choose $\psi$ such that e.g. $\psi(x_1) = 0$ and $\psi(x_2) = 1$. Thus $\Psi(u_1, x_1) = \psi(x_1) \neq \psi(x_2) = \Psi(u_2, x_2)$.

**Case 2: $x_1 = x_2 = x^*$ and $u_1 \neq u_2$.** This is the more difficult case. In the following argument, we will appeal to the universality of shallow neural networks, and their compositions, multiple times. We will forego the tedious $\epsilon$-$\delta$ estimates, and instead sketch out the general idea; filling in the details is a straightforward exercise that would not provide any additional insight.

The underlying idea is that – roughly by linearization of the hidden layers of the underlying non-linear convolutional architecture – we can construct $\Psi$ which approximately takes the form,

$$\Psi(u)(x) = \int_D \bar{g}(x - y) R_1(u(y), y) \, dy$$
$$= \int_D (g_L * g_{L-1} * \cdots * g_1)(x - y) R_1(u(y), y) \, dy$$

where $R_1$ is an arbitrary multilayer perceptron. Given this simplified form, and since $\bar{g}$ has full support by assumption, it turns out that we can always construct suitable $R_1$ that ensures that the we can separate the input functions $u_1$ and $u_2$, ultimately ensuring that $\Psi(u_1)(x^*) \neq \Psi(u_2)(x^*)$. We provide the details of the construction below, starting with the "linearization trick".

**Linearization trick:** We construct $\Psi$ as follows: First, given matrices $A_\ell$ and functions $g_\ell$ as in the (rigorous) definition of "full field of view", we choose the weights of the hidden layers $\mathcal{L}_\ell$, such that

$$\mathcal{L}_\ell(v)(x) = \sigma\left(A_\ell \int_D g_\ell(x - y) v(y) \, dy + b_\ell\right).$$

This is possible by assumption on the convolutional NO architecture, cp. equation 8. By universality of shallow neural networks, we can choose $H$ sufficiently large, and choose matrices $C_*, A_*$ and biases $\alpha_*, \beta_*$, such that for all relevant input vectors $\xi = (\xi_1, 0, \dots, 0)$ (effectively one-dimensional), we have

$$C_* \sigma(A_* \xi + \beta_*) + \alpha_* \approx \xi, \tag{10}$$

i.e. the resulting shallow neural network is an approximation of the identity on one-dimensional $\xi$, to any desired accuracy.

We now momentarily focus on the case $L = 2$. In this case, we choose $A_1 = A_*$, $b_1 = b_*$, then choose $A_2 = A_* C_*$, and bias $b_2 = \alpha_* \int_D g(x - y) \, dy + \beta_*$, so that

$$\mathcal{L}_2 \circ \mathcal{L}_1(v)(x) = \sigma\left(A_2 \int_D g_2(x - y) \mathcal{L}_1(v)(y) \, dy + b_2\right),$$

where

$$A_2 \int_D g_2(x - y) \mathcal{L}_1(v)(y) \, dy + b_2 = A_* \int_D g_2(x - y) \{C_* \mathcal{L}_1(v)(y) + \alpha_*\} \, dy + b_*.$$

We now note that, by construction, we have for any hidden state function of the form $v(y) = [v_1(y), 0, \dots, 0]^T$:

$$C_* \mathcal{L}_1(v)(y) + \alpha_* = C_* \sigma(A_*(g_1 * v)(y) + \beta_*) + \alpha_* \approx \int_D g_1(y - z) v(z) \, dz,$$

where we have used equation 10 and note that this approximation is to any desired accuracy. It follows that also

$$\mathcal{L}_2 \circ \mathcal{L}_1(v)(x) \approx \sigma \left( A_* \int_D g_2(x-y) \int_D g_1(y-z)v(z)\,dz\,dy + b_* \right)$$
$$= \sigma \left( A_* \int_D (g_2 * g_1)(x-y)v(y)\,dy + b_* \right),$$

to any desired accuracy. This shows that for two hidden layers $\mathcal{L}_1, \mathcal{L}_2$ there exists a choice of the parameters, such that for a hidden state $v(x) = (v_1(x), 0, \dots, 0)$, we can obtain an arbitrarily good approximation

$$\mathcal{L}_2 \circ \mathcal{L}_1(v)(x) \approx \sigma \left( A_* \int_D (g_2 * g_1)(x-y)v(y)\,dy + b_* \right).$$

Iterating this argument, for general $L \geq 2$, we start from $\mathcal{L}_L \circ \mathcal{L}_{L-1} \circ \dots \mathcal{L}_1$, and we first choose the weights of $\mathcal{L}_L$ and $\mathcal{L}_{L-1}$ such that

$$\mathcal{L}_L \circ \mathcal{L}_{L-1}(v)(x) \approx \sigma \left( A_* \int_D (g_L * g_{L-1})(x-y)v(y)\,dy + b_* \right).$$

This can be done by the argument employed for the case $L = 2$. Next, we can apply the same argument again to choose the weights of $\mathcal{L}_{L-2}$, such that

$$\mathcal{L}_L \circ \mathcal{L}_{L-1} \circ \mathcal{L}_{L-2}(v)(x) = (\mathcal{L}_L \circ \mathcal{L}_{L-1}) \circ \mathcal{L}_{L-2}(v)(x)$$
$$\approx \sigma \left( A_* \int_D (g_L * g_{L-1} * g_{L-2})(x-y)v(y)\,dy + b_* \right),$$

and continue similarly, until

$$\mathcal{L}_L \circ \dots \circ \mathcal{L}_1(v)(x) \approx \sigma \left( A_* \int_D (g_L * \dots * g_1)(x-y)v(y)\,dy + b_* \right),$$

to any desired accuracy. This argument is based on the assumption that $v(x) = (v_1(x), 0, \dots, 0)$.

We next add a projection layer $\mathcal{Q}$ to this composition, of the form $\mathcal{Q}(v)(x) = Q(C_* v(x) + \alpha_*)$, where $Q$ is a shallow neural network, to obtain

$$\mathcal{Q} \circ \mathcal{L}_L \circ \dots \circ \mathcal{L}_1(v)(x) \approx Q \left( \int_D \bar{g}(x-y)v(y)\,dy \right), \quad \bar{g}(x) := (g_L * \dots * g_1)(x).$$

Recall that the support of $(x, y) \mapsto \bar{g}$ is $D \times D$, which, roughly speaking, might be thought of as $\bar{g}(x-y) \neq 0$ for all $x, y \in D$. The non-vanishing support is by assumption (full field of view).

Pre-composing with a lifting layer $\mathcal{R}(u)(x) = R(u(x), x)$, which we choose to have only a non-vanishing first component on the output-side, i.e. $R(u(x), x) = (R_1(u(x), x), 0, \dots, 0)$, it follows that we can construct $\Psi(u)(x) := \mathcal{Q} \circ \mathcal{L}_L \circ \dots \circ \mathcal{L}_1 \circ \mathcal{R}$, such that

$$\Psi(u)(x) \approx Q \left( \int_D \bar{g}(x-y) R_1(u(y), y)\,dy \right),$$

where the approximation error can be made arbitrarily small. Here, we are still free to choose $Q$ and $R_1$.

Our goal is to choose them in such a way that for our given functions $u_1 \neq u_2$ and the given point $x^* = x_1 = x_2 \in D$, we have $\Psi(u_1)(x_1) = \Psi(u_1)(x^*) \neq \Psi(u_2)(x^*) = \Psi(u_2)(x_2)$. We will choose $Q$ as an approximation of the identity on the first component, so that

$$\Psi(u)(x) \approx \int_D \bar{g}(x-y) R_1(u(y), y)\,dy.$$

**Separating** $(u_1, x_1)$ **from** $(u_2, x_2)$ **when** $u_1 \neq u_2$, $x_1 = x_2 = x^*$: By assumption, $\bar{g}(x^* - y) \neq 0$ has full support for $y \in D$. Since $u_1 \neq u_2$, there exists $y_0 \in D$ such that $u_1(y_0) \neq u_2(y_0)$. We may wlog assume that $u_1(y_0) < \tau_1 < \tau_2 < u_2(y_0)$ for some $\tau_1, \tau_2 \in \mathbb{R}$. By continuity of $u_1, u_2$, there exists $\delta > 0$, such that $\max_{B_\delta(y_0)} u_1(y) < \tau_1 < \tau_2 < \min_{B_\delta(y_0)} u_2(y_0)$. By the universality of the shallow network $R_1$, we can choose $R_1$, such that we approximately have

$$R_1(\eta, y) \approx \rho_\delta(y - y_0) h_{\tau_1, \tau_2}(\eta),$$

where $\rho_\delta(\,\cdot\, - y_0)$ is a non-negative function supported inside $B_\delta(y_0)$, and $h_{\tau_1, \tau_2}(\,\cdot\,)$ is a non-negative function, such that

$$h_{\tau_1, \tau_2}(\eta) = \begin{cases} 1, & (\eta > \tau_2), \\ 0, & (\eta < \tau_1). \end{cases}$$

Since $y \mapsto \bar{g}(x^* - y)$ has full support, and since $\rho_\delta(y - y_0)$ is an arbitrary function apart from the constraints we imposed above, we can further refine our choice of $\rho_\delta$, to ensure that

$$\int_D \bar{g}(x^* - y) \rho_\delta(y - y_0) \, dy \neq 0.$$

It then follows that

$$R_1(u_1(y), y) \approx \rho_\delta(y - y_0) h_{\tau_1, \tau_2}(u_1(y)) \equiv 0,$$

and

$$R_1(u_2(y), y) \approx \rho_\delta(y - y_0) h_{\tau_1, \tau_2}(u_2(y)) \equiv \rho_\delta(y - y_0).$$

In particular, we conclude that – to any desired accuracy – we can construct $\Psi(u)$, such that

$$\Psi(u_1)(x^*) \approx \int_D \bar{g}(x^* - y) R_1(u_1(y), y) \, dy = 0$$

and

$$\Psi(u_2)(x^*) \approx \int_D \bar{g}(x^* - y) R_1(u_2(y), y) \, dy \approx \int_D \bar{g}(x^* - y) \rho_\delta(y - y_0) \, dy \neq 0.$$

In particular, given the above two approximate identities, upon making the approximation errors sufficiently small, it follows that there exists $\Psi \in \mathbb{A}$ such that $\Psi(u_1)(x_1) = \Psi(u_1)(x^*) \neq \Psi(u_2)(x^*) = \Psi(u_2)(x_2)$. This finally shows that $\mathbb{A}$ separates points inputs $(u_1, x_1)$ and $(u_2, x_2)$, under the assumption that $u_1 \neq u_2$ and $x_1 = x_2 = x^*$, and concludes our proof for **Case 2**.

Our proof of the universality of $\mathbb{A} \subset C(\mathcal{K} \times D)$ now concludes by application of the Stone-Weierstrass theorem (cp. Lemma B.2).

$\qquad\qquad\qquad\qquad\qquad\qquad\qquad\qquad\qquad\qquad\qquad\qquad\qquad\qquad\qquad\qquad\qquad\qquad\qquad\qquad\qquad$ $\square$

## C   Further Ablation Studies

### C.1   Unidirectional Spatial SSM

We conduct an ablation study to evaluate the importance of the bidirectional spatial module in our SS-NO architecture on 1D Burgers' Equation. To ensure a fair comparison that isolates the effect of directionality, we construct a unidirectional model with the same parameter count as the full SS-NO. This is done by stacking 8 layers of forward-only spatial SSM blocks and inserting a single temporal SSM layer in the middle.

As shown in Figure 7, the unidirectional variant fails to capture the dynamics effectively. Its training curve plateaus early, and the relative $\ell_2$ error stagnates around 0.075. In contrast, the full bidirectional model continues to improve and ultimately reaches a much lower error of approximately 0.007—more than ten times better. These results highlight that unidirectional spatial SSMs are fundamentally limited in their representational capacity for 2D PDEs and act as non-universal approximators in this setting – indeed, this unidirectional spatial SSM *is lacking* a full field of view, violating the criterion for universality in Theorem 4.1. Bidirectional context is essential to capture the long-range spatial dependencies needed for accurate forecasting.

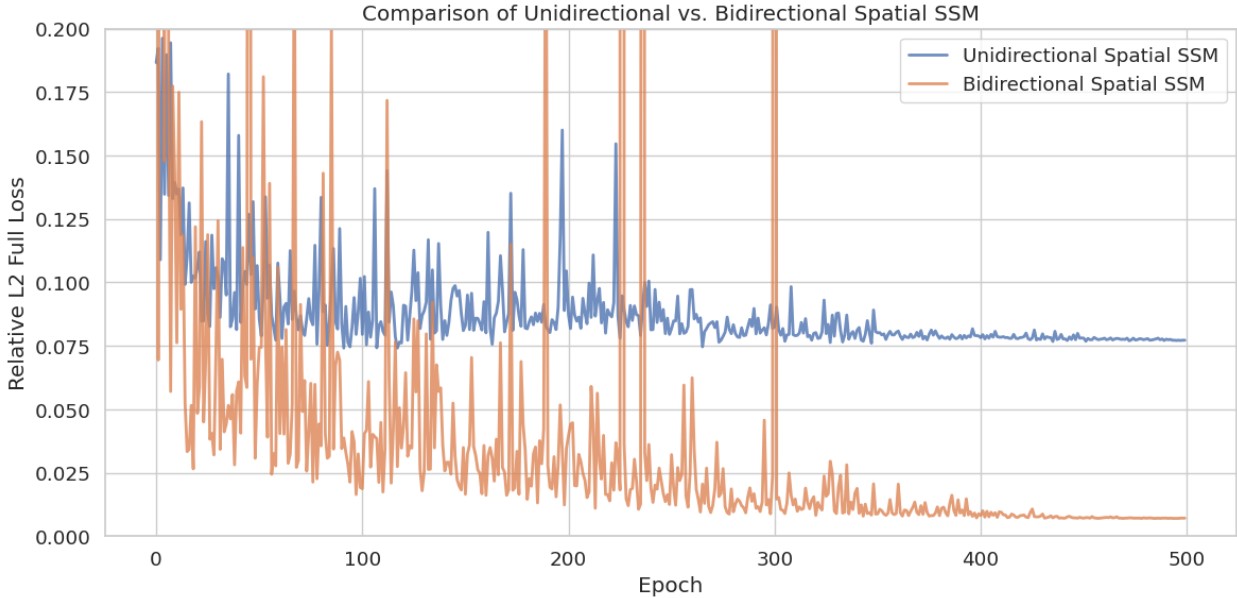

Figure 7: Training curves comparing the unidirectional and bidirectional spatial SSMs on the `TorusLi` dataset. The unidirectional variant plateaus early, failing to capture global spatial dependencies.

## C.2 Effect of Missing Contextual Information and the Role of Temporal Modeling

To better understand the role of temporal modeling in our architecture, we evaluate the full SS-NO against an ablated variant that removes the temporal state-space module (S4), as well as two strong baselines (FFNO and FNO2D). Figure 8 shows the relative $\ell_2$ error across three levels of contextual information: *all context*, *only forcing*, and *no context*, on both the `TorusVis` and `TorusVisForce` datasets.

Three main observations emerge from these results. First, the temporal S4 component is necessary in the most challenging *No Context* setting, where both viscosity and forcing inputs are missing. Without temporal modeling, SS-NO suffers from very high errors (e.g., 0.1156 vs. 0.0411 on `TorusVis`), whereas the full model remains comparatively robust by leveraging temporal dependencies to compensate for missing information. These results extend the observations made by Buitrago et al. (2025), who showed that temporal memory helps mitigate the effects of resolution loss and noisy data. In our case, we show that memory-based modeling also helps compensate for missing physical parameters—highlighting an additional benefit of incorporating temporal memory into spatiotemporal PDE models.

Second, in the *Only Force* setting, the presence of forcing information appears to be sufficient for accurate predictions in our architecture: both the full and ablated SS-NO achieve consistently low errors, while FFNO exhibits a slight degradation. This suggests that temporal modeling is not strictly necessary when forcing is available, as even the ablated variant performs competitively.

Finally, in the extreme case where all context is missing, FFNO surprisingly outperforms SS-NO (full), despite performing worse in other settings. We hypothesize this could be the result of the difference between how FFNO and FNO/SS-NO make residual connections, since it seems like the degree of performance loss is similar between SS-NO (full) and FNO2D, but we do not do further investigation as it is out of the scope of this work.

## C.3 Effect of Varying the Memory Window Size

We investigate how the choice of memory window size $K$ affects the performance of our SS-NO model on the KS dataset with $\nu = 0.075$. This hyperparameter controls how many past spatial feature maps are accessible

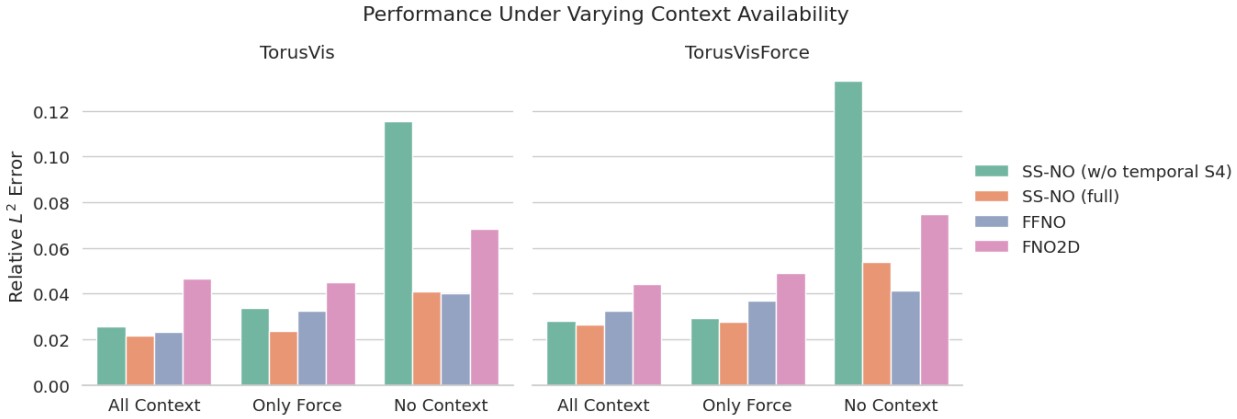

Figure 8: Relative $\ell_2$ error across three levels of contextual information: full context, only forcing, and no context. Lower is better.

to the model during temporal state updates. All results reported in this section are obtained from models trained *without teacher forcing* to better reflect deployment conditions.

As shown in Figure 9, increasing $K$ consistently improves performance up to around $K = 8$. Beyond this point, the performance gains become increasingly marginal, and the validation loss begins to plateau. Notably, setting $K = 0$, which corresponds to no temporal memory, results in significantly degraded performance. These results highlight the critical importance of temporal memory in modeling complex spatiotemporal dynamics, while also suggesting diminishing returns beyond a moderate window size.

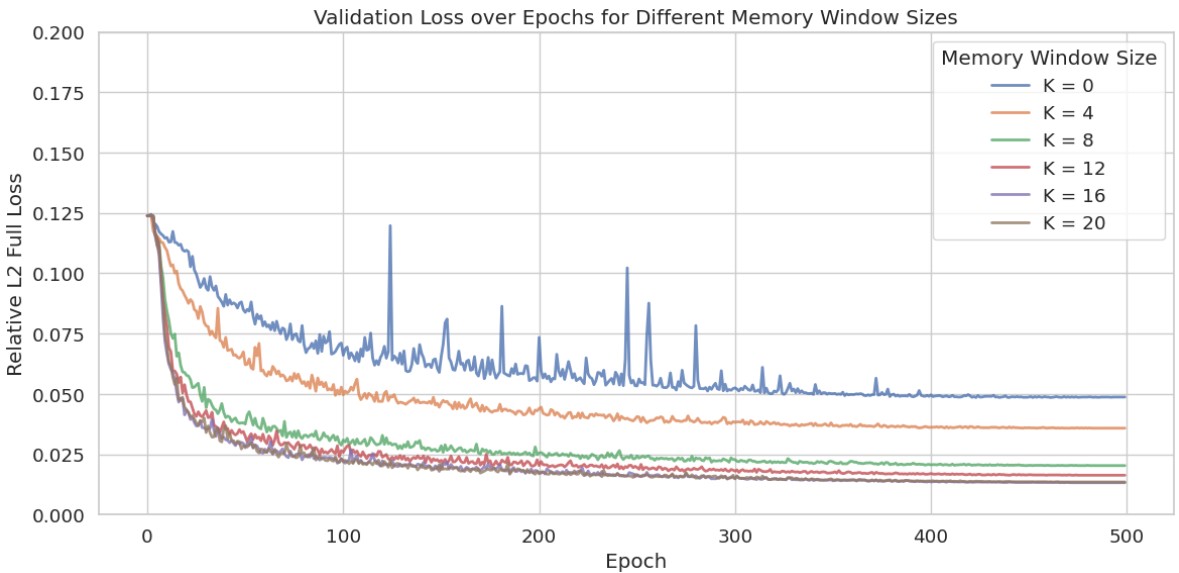

Figure 9: Validation $\ell_2$ loss over training epochs for different memory window sizes $K$ on the KS dataset ($\nu = 0.075$). Increasing the window improves performance until around $K = 8$, after which gains diminish.

## C.4 Effect of Teacher Forcing on SS-NO Performance

**Case Study: 1D KS with $\nu = 0.075$ at Low Resolution.** While prior work such as FFNO (Tran et al., 2023) suggests that teacher forcing can improve model performance by stabilizing training, we observe a different trend for our SS-NO architecture, particularly in low-resolution regimes. To investigate this, we

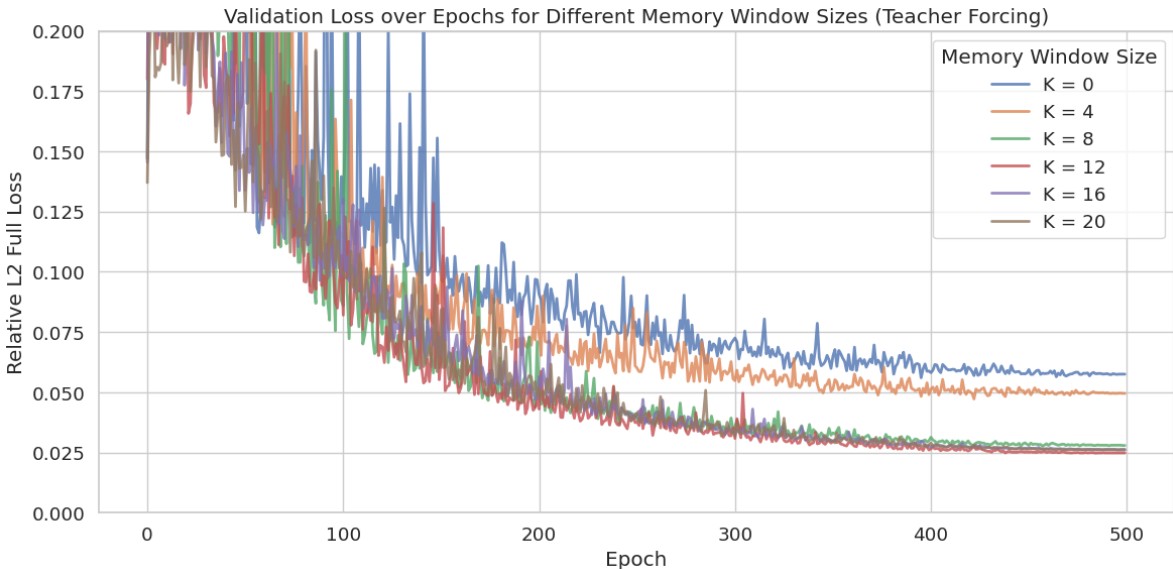

Figure 10: Validation $\ell_2$ loss over training epochs on the KS dataset ($\nu = 0.075$) using teacher forcing for various memory window sizes $K$. Compared to Figure 9, training is less stable and results in higher loss.

revisit the 1D Kuramoto–Sivashinsky (KS) dataset with $\nu = 0.075$, where reduced spatial resolution poses a significant challenge.

We train SS-NO with teacher forcing across various memory window sizes ($K = 0$ to 20), and present the results in Figure 10. Compared to the non-teacher-forced variant (Figure 9), training with teacher forcing leads to noticeably less stable optimization, slower convergence, and consistently higher final relative $\ell_2$ errors. Moreover, while performance still improves with longer memory, the gains plateau earlier, and the benefits beyond $K = 8$ diminish more rapidly than in the non-teacher-forced case.

These findings suggest that in this setting, SS-NO benefits more from fully autoregressive training, likely due to its strong inductive bias for modeling long-term temporal dependencies without relying on ground-truth guidance.

**General Trends Across 1D and 2D Benchmarks.** To better understand the broader impact of teacher forcing, we compare SS-NO models trained with and without teacher forcing across all 1D and 2D benchmarks (Tables 3 and 4).

In 1D settings, results are mixed: at higher resolutions (e.g., 128), teacher forcing slightly improves performance in some cases (e.g., KS with $\nu = 0.1$ and $\nu = 0.125$), but at lower resolutions, non-teacher-forced models generally perform better—especially in the challenging $\nu = 0.075$ KS setup. These results suggest that autoregressive training may confer more robustness in low-resolution or harder regimes, where model predictions diverge more easily from ground truth.

In contrast, across all 2D Navier–Stokes benchmarks, teacher forcing consistently yields better performance. This suggests that for higher-dimensional systems with complex spatiotemporal interactions, teacher forcing can help stabilize training and guide the model toward more accurate trajectories.

Overall, while the effect of teacher forcing appears to be dataset- and resolution-dependent, we find that non-teacher-forced training can be highly effective in 1D regimes, particularly under low resolution. In contrast, for complex 2D flows, teacher forcing remains a valuable tool for improving predictive accuracy.

Table 3: Relative $\ell_2$ error on 1D benchmarks at varying resolutions.

| Resolution | Architecture | Relative $\ell_2$ Error | | | |
|---|---|---|---|---|---|
| | | KS | | | Burgers' |
| | | $\nu = 0.075$ | $\nu = 0.1$ | $\nu = 0.125$ | $\nu = 0.001$ |
| 128 | SS-NO (Teacher Forcing) | 0.0086 | **0.0027** | **0.0013** | **0.0070** |
| | SS-NO (No Teacher Forcing) | **0.0076** | 0.0034 | 0.0019 | **0.0070** |
| 64 | SS-NO (Teacher Forcing) | 0.0143 | 0.0048 | 0.0029 | 0.0121 |
| | SS-NO (No Teacher Forcing) | **0.0135** | **0.0042** | **0.0023** | **0.0114** |
| 32 | SS-NO (Teacher Forcing) | 0.0472 | 0.0162 | 0.0088 | 0.0200 |
| | SS-NO (No Teacher Forcing) | **0.0358** | **0.0135** | **0.0071** | **0.0171** |

Table 4: Relative $\ell_2$ error on 2D Navier–Stokes datasets. All models are evaluated at a spatial resolution of $64 \times 64$.

| Architecture | Relative $\ell_2$ Error | | |
|---|---|---|---|
| | TorusLi | TorusVis | TorusVisForce |
| SS-NO (Teacher Forcing) | **0.0345** | **0.0218** | **0.0263** |
| SS-NO (No Teacher Forcing) | 0.0546 | 0.0385 | 0.0371 |

## D Comparison of Factorized Architectural Variants

In this section, we explore how different architectural factorization strategies affect performance on the `TorusLi` dataset. Specifically, we examine two popular alternatives to our default SS-NO spatial block.

**SS-NO-VM (Vision Mamba style).** This variant adapts the Vision Mamba architecture by replacing the Mamba core with our spatial SSM block. The key idea is to preserve Vision Mamba's flattened zigzag scanning structure in 2D, allowing bidirectional processing over a 1D sequence obtained from flattening the spatial grid. After bidirectional processing over the first spatial axis, the result is passed through a mixing linear projection before a second pass along the alternate axis, again using bidirectional spatial SSMs and linear mixing. This model maintains Vision Mamba's emphasis on alternating directional fusion while leveraging the structure of our SSM.

**SS-NO-FF (FFNO style).** In this variant, we discard the original spatial block of SS-NO and instead adopt the FFNO-style connection, where spatial context is aggregated through two 1D sweeps: one across the $x$-axis and one across the $y$-axis. Each sweep involves a forward and backward spatial SSM applied independently along that axis, and the outputs from both axes are summed together. This approach highlights the role of directional decoupling in factorized Fourier models and contrasts with the fused zigzag sweep in SS-NO-VM.

Pseudocode for both SS-NO-FF and SS-NO-VM variants is provided in Figure 11.

**FNO2D-R (Reduced Fourier Kernel).** To verify that the gains of SS-NO are not trivially due to kernel simplification, we construct a reduced FNO2D model, denoted as FNO2D-R. In this variant, we remove the output channel mixing in the Fourier kernel by modifying the kernel from:

```
self.weights = nn.Parameter(self.scale * torch.rand(
    in_channels, out_channels, modes1, modes2, dtype=torch.cfloat))
...
return torch.einsum("bixy,ioxy->boxy", input, weights)
```

```
# SS-NO-VM (Vision Mamba Style)
x_flat = flatten_spatial(x)

# First zigzag sweep
x_fwd = SSM_flat_forward(x_flat)
x_bwd = SSM_flat_backward(reverse(x_flat))
z1 = permute_channels(x_fwd + x_bwd)
x = linearT(z1) + x_flat

# Second zigzag sweep
x_fwd2 = SSM_flat_forward(x)
x_bwd2 = SSM_flat_backward(reverse(x))
z2 = permute_channels(x_fwd2 + x_bwd2)
x = linearT2(z2) + x

x_out = reshape_spatial(x)
```

Listing 1: SS-NO-VM (Vision Mamba style)

```
# SS-NO-FF (FFNO-style)
residual = x

# X-axis sweep
x1 = reshape_for_x_sweep(x)
x1_fwd = SSM_x_forward(x1)
x1_bwd = reverse(SSM_x_backward(reverse(x1)))
x1_combined = reshape_back(x1_fwd + x1_bwd)

# Y-axis sweep
y1 = reshape_for_y_sweep(x)
y1_fwd = SSM_y_forward(y1)
y1_bwd = reverse(SSM_y_backward(reverse(y1)))
y1_combined = reshape_back(y1_fwd + y1_bwd)

# Combine and project
z = x1_combined + y1_combined
z = backcast_ff(z)
output = z + residual
```

Listing 2: SS-NO-FF (FFNO-style connection)

Figure 11: Pseudocode for two spatial architectural variants of SS-NO: Vision Mamba-style (left) and FFNO-style (right).

to:

```
self.weights = nn.Parameter(self.scale * torch.rand(
    in_channels, modes1, modes2, dtype=torch.cfloat))
...
return input * weights
```

This isolates the contribution of full channel-wise Fourier mixing and allows for a more controlled comparison against SS-NO.

As shown in Table 5, our original SS-NO achieves the best performance among all tested variants, demonstrating that both our architectural choices and structured state-space modeling contribute meaningfully to the observed improvements.

Table 5: Relative $\ell_2$ error on the `TorusLi` dataset for different factorized architectural variants. All models are evaluated at a spatial resolution of $64 \times 64$.

| Architecture | # Parameters | Relative $\ell_2$ Error |
|---|---|---|
| | | TorusLi |
| SS-NO-VM | 402,945 | 0.0495 |
| SS-NO-FF | 503,297 | 0.0403 |
| FNO2D-R | $1,115,841$ | 0.0718 |
| SS-NO (ours) | 369,665 | **0.0345** |

# E  Data Preprocessing and Training Details

## E.1  Data Preprocessing and Training Setup

We normalize all input data to the range $[0, 1]$ and add fixed-variance Gaussian noise during training. Through empirical validation, we found $\sigma = 0.005$ optimal for Burgers' equation to stabilize training while preserving signal integrity, while $\sigma = 0.001$ works best for all other datasets. All models are trained for 500 epochs using

the AdamW optimizer with an initial learning rate of $10^{-3}$, weight decay of $10^{-4}$, and a cosine annealing learning rate schedule. We employ teacher forcing during training unless otherwise noted.

To simulate low-resolution scenarios, we generate downsampled versions of the original datasets by uniformly subsampling in space. All models are trained to minimize the normalized step-wise relative $\ell_2$ loss, and evaluation is reported using the full-sequence relative $\ell_2$ error.

## E.2 Model Architecture Specifications

All models use four blocks with a consistent hidden dimension of 64 across both 1D and 2D problems. Intermediate and output linear layers maintain a hidden size of 128. With the exception of FFNO and our SS-NO on 1D problems, we adopt the hyperparameter settings reported by Buitrago et al. (2025). Unlike their setup which uses hidden dimension 128 for 1D and 64 for 2D problems, we found no statistical difference in performance with a unified hidden dimension of 64, simplifying architecture design while maintaining competitive results.

All models except GKT follow the MemNO framework and incorporate a temporal S4 module with window size $K = 4$ positioned in the middle of the network stack. For GKT, we use a multi-input variant where the temporal dimension is mixed with features.

We adopt a simple spatial positional encoding scheme shared across all models. In 1D, for a grid with $f$ equispaced points over the interval $[0, L]$, the positional encoding $E \in \mathbb{R}^f$ is defined as $E_i = \frac{i}{L}$ for $0 \leq i < f$. In 2D, for a $f \times f$ grid over $[0, L_x] \times [0, L_y]$, the encoding is $E_{ij} = (\frac{i}{L_x}, \frac{j}{L_y})$. The input lifting operator $r_{\text{in}}$ is a shared linear layer that maps the concatenation of input features and positional encoding to the hidden space, applied pointwise across the spatial grid.

## E.3 Baseline Model Architectures

Unless otherwise specified, all models use four blocks with a channel width of 64. Intermediate and output linear layers have a hidden size of 128 (i.e., twice the channel width). **With the exception of FFNO, multi-input FNO, and our SS-NO on 1D problems, we adopt the "optimal" hyperparameter settings (i.e., number of layers, hidden dimension, and expansion size of linear layers) reported by Buitrago et al. (2025).** The U-Net we use is even larger, containing more parameters than this standard baseline. For memory-augmented architectures, the memory module is inserted after the first two blocks and before the last two. For all non-Markovian models, we fix the memory window size to $K = 4$ across experiments; the impact of varying $K$ on SS-NO is explored in Appendix C.3.

We adopt a simple spatial positional encoding scheme shared across all models. In 1D, for a grid with $f$ equispaced points over the interval $[0, L]$, the positional encoding $E \in \mathbb{R}^f$ is defined as $E_i = \frac{i}{L}$ for $0 \leq i < f$. In 2D, for a $f \times f$ grid over $[0, L_x] \times [0, L_y]$, the encoding is $E_{ij} = (\frac{i}{L_x}, \frac{j}{L_y})$. The input lifting operator $r_{\text{in}}$ is a shared linear layer that maps the concatenation of input features and positional encoding from $\mathbb{R}^{2+k}$, where $k$ is the number of features, to the hidden space $\mathbb{R}^h$, applied pointwise across the spatial grid. Similarly, a shared decoder $r_{\text{out}}$ maps from $\mathbb{R}^h$ back to $\mathbb{R}$. For all models containing an FNO or FFNO module, we follow the setup in Buitrago et al. (2025) and retain all available Fourier modes by setting $k_{\max} = \lfloor \frac{f}{2} \rfloor$, where $f$ denotes the spatial resolution. A notable exception is the `TorusLi` dataset, where an 8-mode FNO configuration outperformed higher mode counts (16, 24, and 32). We therefore present the results based on 8 modes for this specific case.

**Factorized Fourier Neural Operator (FFNO).** The Factorized Fourier Neural Operator (FFNO), introduced by Tran et al. (2023), extends the original Fourier Neural Operator (FNO) (Li et al., 2021) by modifying how the spectral integral kernel is applied. We use the standard FFNO implementation from `https://github.com/alasdairtran/fourierflow/`. Each FFNO layer operates on a spatial grid of size $|S|$ and a hidden dimension $h$, and is defined as a residual block:

$$\ell(v) = v + \text{Linear}_{h \to h'} \circ \sigma \circ \text{Linear}_{h' \to h} \circ \mathcal{K}(v),$$

where $\sigma$ denotes the ReLU activation function (Nair & Hinton, 2010), and $h'$ is an intermediate dimensionality used within the nonlinear mapping. The integral operator $\mathcal{K}$ transforms $v$ in the Fourier domain and is computed by aggregating across spatial dimensions:

$$\mathcal{K}(v) = \sum_{\alpha=1}^{d} \mathrm{IFFT}_\alpha \left[ R_\alpha \cdot \mathrm{FFT}_\alpha(v) \right],$$

where $\mathrm{FFT}_\alpha$ and $\mathrm{IFFT}_\alpha$ are the (inverse) discrete Fourier transforms applied along the $\alpha$-th spatial axis, and $R_\alpha \in \mathbb{C}^{h^2 \times k_{\max}}$ are learned complex-valued projection matrices for each dimension.

**Fourier Neural Operator (FNO).** For 2D problems, we use a modified version of the original Fourier Neural Operator (FNO) (Li et al., 2021) where the output of the integral kernel is passed through a nonlinearity before the residual skip connection, following the implementation at `https://github.com/lilux618/fourier_neural_operator/blob/master/fourier_2d_time.py`, which we found to perform better.

**Galerkin Transformer (GKT).** We use the Galerkin Transformer (GKT) implementation from `https://github.com/scaomath/galerkin-transformer`. We employ a multi-input variant where the model is made non-Markovian by providing the last $K = 4$ steps as input and processing the temporal dimension as normal channels concatenated with other features, rather than as an independent dimension. We use a hidden size of 32 with dropout rates of 0.05 in attention layers and 0.025 in feedforward layers.

**Factformer 2D.** As part of our evaluation, we include the original Factformer model from Li et al. (2023a), which is designed for spatiotemporal modeling over 2D spatial domains. While Buitrago et al. (2025) only evaluated the 1D variant, we adopt the same architectural configuration for the 2D case to ensure fair comparison: four attention layers, a hidden dimension of 64, and 4 attention heads with a total projection dimension of 512. Factformer 2D applies linear attention sequentially along each spatial axis. Given a hidden tensor $w \in \mathbb{R}^{S_x \times S_y \times H}$, two separate MLPs ($\mathrm{MLP}_x$ and $\mathrm{MLP}_y$) are applied to compute keys and queries across $S_x$ and $S_y$, respectively. After averaging over the complementary axis, we obtain $q_x, k_x \in \mathbb{R}^{S_x \times H}$ and $q_y, k_y \in \mathbb{R}^{S_y \times H}$. Values $v \in \mathbb{R}^{S_x \times S_y \times H}$ are computed through a shared linear projection, and attention is applied first along $x$ and then $y$. Our implementation follows the original GitHub repository `https://github.com/BaratiLab/FactFormer` with minimal changes limited to data format compatibility.

**Factformer 1D.** We also evaluate the 1D version of Factformer, as introduced in Buitrago et al. (2025), using the same configuration of four attention layers, hidden dimension 64, and 4 attention heads (total projection dimension 512). Unlike the 2D case, the 1D variant processes inputs over a single spatial dimension. As such, only one MLP ($\mathrm{MLP}_x$) is used to compute queries and keys, and no spatial averaging is applied. A single linear attention operation is performed per layer along the 1D spatial axis. Values are projected from the input using a linear layer. This model is implemented within the same codebase as the 2D version to maintain consistency, with minor modifications to handle 1D inputs.

**U-Net Neural Operator (U-Net).** Our U-Net implementation follows the typical encoder-decoder architecture with skip connections (Gupta & Brandstetter, 2023). It consists of four downsampling convolutional blocks, a bottleneck convolutional block, and four upsampling blocks with residual connections linking corresponding encoder and decoder layers. We use a first hidden dimension of 32, and apply channel multipliers of $[1, 2, 4, 8]$ in the encoder path. No time embeddings are used. This setup is adapted to process spatiotemporal data by flattening the spatial dimensions and independently applying the network to each time step. Our implementation is based on the publicly available codebase from PDEBench `https://github.com/pdebench/PDEBench` (Takamoto et al., 2022). For reference, the U-Net Neural Operator variant used in prior work by Buitrago et al. (2025) employs a similar architecture but with channel multipliers of $[1, 2, 2, 2]$.

## F   Data Generation

### F.1   1D Burgers' Equation

We consider the one-dimensional viscous Burgers' equation, expressed as:

$$\partial_t u + u\,\partial_x u = \nu\,\partial_{xx}u, \quad (t,x) \in [0,T] \times [0,L],$$

where $\nu$ is the viscosity coefficient. For our experiments, we utilize the publicly available 1D Burgers' dataset from PDEBench (Takamoto et al., 2022), with viscosity $\nu = 0.001$. The original dataset contains 10,000 spatiotemporal samples, each defined on a spatial grid of $1,024$ points and 201 time steps over the interval $[0, 2.01]$ seconds. We restrict our usage to the first 140 time steps and uniformly downsample them to obtain 20 steps spanning up to 1.4 seconds. This truncation is motivated by the observation that, after this point, the dissipative effect of the diffusion term $\partial_{xx}u$ suppresses high-frequency dynamics, causing the solution to evolve slowly (Buitrago et al., 2025). For training, we use the first 2,048 samples from the dataset and reserve the last 1,000 samples for testing.

### F.2   1D Kuramoto–Sivashinsky Equation

The Kuramoto–Sivashinsky (KS) equation is given by:

$$\partial_t u + u\,\partial_x u + \partial_{xx}u + \nu\,\partial_{xxxx}u = 0, \quad (t,x) \in [0,T] \times [0,L],$$

with periodic boundary conditions. We follow the exact same setting as provided by Buitrago et al. (2025), using their public repository at `https://github.com/r-buitrago/LPSDA`, which builds upon the implementation of Brandstetter et al. (2022). The spatial domain $[0, 64]$ is discretized into 512 points, and the temporal domain $[0, 2.5]$ is divided into 26 equispaced time steps with fixed $\Delta t = 0.1$.

Initial conditions are generated as random superpositions of sine waves:

$$u_0(x) = \sum_{i=0}^{20} A_i \sin\left(\frac{2\pi k_i}{L}x + \phi_i\right),$$

where for each trajectory, the amplitudes $A_i$ are sampled from a continuous uniform distribution on $[-0.5, 0.5]$, the frequencies $k_i$ are drawn from a discrete uniform distribution over $\{1, 2, \ldots, 8\}$, and the phases $\phi_i$ are sampled uniformly from $[0, 2\pi]$.

We consider three different viscosities: $\nu = 0.075$, 0.1, and 0.125. For each viscosity, we generate $2,048$ training samples and 256 validation samples.

### F.3   2D Navier Stokes Equations

**The `TorusLi` Dataset.**   We consider the two-dimensional incompressible Navier–Stokes equations on the unit torus $\mathbb{T}^2 = [0,1]^2$ in vorticity form. The evolution of the scalar vorticity field $\omega(x,y,t)$ is governed by:

$$\partial_t \omega + \mathbf{u} \cdot \nabla \omega = \nu \Delta \omega + f,$$

where $\mathbf{u} = (u,v)$ is the velocity field satisfying the incompressibility condition $\nabla \cdot \mathbf{u} = 0$, $\nu > 0$ is the kinematic viscosity, and $f$ is a fixed external forcing function.

We directly reuse the dataset released by Li et al. (2021), referred to as `TorusLi`, which was originally developed to benchmark the FNO. The simulations were generated using a pseudospectral Crank–Nicolson second-order time-stepping scheme on a high-resolution $256 \times 256$ grid, and subsequently downsampled to $64 \times 64$. All trajectories use a constant viscosity of $\nu = 10^{-5}$ (corresponding to a Reynolds number Re = 2000), and share the same external forcing:

$$f(x,y) = 0.1 \left[\sin(2\pi(x+y)) + \cos(2\pi(x+y))\right].$$

Initial vorticity fields $\omega_0$ are sampled from a Gaussian random field:

$$\omega_0 \sim \mathcal{N}\left(0, 7^{3/2}(-\Delta + 49I)^{-2.5}\right),$$

with periodic boundary conditions. The numerical solver computes the velocity field by solving a Poisson equation in Fourier space and evaluates nonlinear terms in physical space with dealiasing applied. The nonlinear term is handled explicitly in the Crank–Nicolson scheme.

The dataset consists of solutions recorded every $t = 1$ time unit, with a total of 20 time steps per trajectory, corresponding to a final time horizon of $T = 20$. This makes the task relatively long-range compared to other PDE benchmarks. The spatial resolution is fixed at $64 \times 64$ for all experiments in this paper.

**The `TorusVis` and `TorusVisForce` Datasets.** We utilize two additional datasets, `TorusVis` and `TorusVisForce`, introduced by Tran et al. (2023), to evaluate model generalization under varying physical regimes. Both datasets are generated using the same Crank–Nicolson pseudospectral solver used in `TorusLi`, maintaining consistency in numerical methodology.

These datasets extend the Navier–Stokes setting by incorporating variability in viscosity and external forcing. Specifically, each trajectory uses a randomly sampled viscosity $\nu$ between $10^{-5}$ and $10^{-4}$. The external forcing function is defined as:

$$f(t, x, y) = 0.1 \sum_{p=1}^{2} \sum_{i=0}^{1} \sum_{j=0}^{1} \left[\alpha_{pij} \sin\left(2\pi p(ix + jy) + \delta t\right) + \beta_{pij} \cos\left(2\pi p(ix + jy) + \delta t\right)\right],$$

where $\alpha_{pij}, \beta_{pij} \sim \mathcal{U}[0,1]$ are sampled independently for each trajectory. The parameter $\delta$ controls the temporal variation in the forcing: for `TorusVis`, $\delta = 0$, resulting in time-invariant forcing; for `TorusVisForce`, $\delta = 0.2$, producing a time-varying force.

As with `TorusLi`, the spatial resolution is fixed at $64 \times 64$, and trajectories consist of 20 time steps sampled every $t = 1$ time unit, yielding a total time horizon of $T = 20$.

### F.4 2D Richtmeyer–Meshkov (CE-RM) Problem

We consider the 2D **Richtmeyer–Meshkov (CE-RM)** benchmark for the compressible Euler equations ($\gamma = 1.4$) on the unit square $[0,1]^2$ with periodic boundary conditions. The initial state contains a high-pressure circular region and a heavy fluid on one side of a perturbed interface (Herde et al., 2024). Specifically, the initial pressure and density are given by

$$p_0(x, y) = \begin{cases} 20, & \sqrt{x^2 + y^2} < 0.1, \\ 1, & \text{otherwise}, \end{cases} \tag{11}$$

$$\rho_0(x, y) = \begin{cases} 2, & |x| < I(x, y, \omega), \\ 1, & \text{otherwise}, \end{cases} \tag{12}$$

with initial velocities $v_x = v_y = 0$. The interface $I(x, y, \omega)$ is perturbed by a random Fourier series:

$$I(x, y, \omega) = 0.25 + \epsilon \sum_{j=1}^{10} a_j(\omega) \sin\left(2\pi((x, y) \cdot (1, 0) + b_j(\omega))\right), \tag{13}$$

where $a_j$ and $b_j$ are independent uniform random amplitudes and phases (with $\sum_j a_j = 1$). We integrate the Euler equations up to time $T = 2$, saving 21 equally spaced snapshots in time.

The publicly released **CE-RM dataset** (Herde et al., 2024) contains 1260 trajectories on a $128 \times 128$ grid, which we spatially downsample by a factor of two to $64 \times 64$. Each snapshot includes five fields (density $\rho$, velocity $v_x, v_y$, pressure $p$, and a passive tracer). In our setup, the passive tracer is assumed available as an input at every time step, while the model predicts only the conserved variables $[\rho, v_x, v_y, p]$. We follow the

dataset split of Herde et al. (2024), using the first 1000 trajectories for training and the last 200 for validation. We further apply temporal downsampling by a factor of 2: we take the solution at times indexed $t = 0, 2, 4, 6$ (the first four snapshots) as input and predict the next 7 steps at indices $8, 10, \ldots, 20$. In other words, given the fields at time steps $t_0, t_2, t_4, t_6$, the model predicts the fields at $t_8, \ldots, t_{20}$.

### F.5 2D Gravitational Rayleigh–Taylor (GCE-RT) Problem

The **GCE-RT** problem adds gravitational forcing to the compressible Euler equations on $[0, 1]^2$ with periodic boundaries. We use the two-dimensional Rayleigh–Taylor setup from astrophysics: a $\gamma = 2$ polytropic equilibrium on a model neutron star is perturbed at a random interface (Herde et al., 2024). In cylindrical symmetry ($r = \sqrt{x^2 + y^2}$), the unperturbed pressure and gravitational potential are

$$p(r) = K_0 \left( \frac{\rho_0 \sin(\alpha r)}{\alpha r} \right)^2, \tag{14}$$

$$\phi(r) = -2K_0 \frac{\rho_0 \sin(\alpha r)}{\alpha r}, \tag{15}$$

with $K_0 = p_0/\rho_0^2$ and $\alpha = \sqrt{4\pi G/(2K_0)}$ (with $G = 1$). The initial density profile is

$$\rho(r) = \sqrt{\frac{K_0}{\tilde{K}(r)}} \frac{\rho_0 \sin(\alpha r)}{\alpha r}, \tag{16}$$

where $\tilde{K}(r) = K_0$ for $r < r_{RT}$ and $\tilde{K}(r) = (1 - A/(1 + A))^2 K_0$ for $r \geq r_{RT}$. The Rayleigh–Taylor interface radius is given by

$$r_{RT}(x, y) = 0.25 \left( 1 + a \cos(\text{atan2}(y, x) + b) \right), \tag{17}$$

with random amplitude $a \in [-1, 1]$ and phase $b \in [-\pi, \pi]$. We also perturb the central density $\rho_0$, pressure $p_0$, and Atwood number $A$ via

$$\rho_0 = 1 + 0.2c, \tag{18}$$

$$p_0 = 1 + 0.2d, \tag{19}$$

$$A = 0.05(1 + 0.2e), \tag{20}$$

with $c, d, e \sim \mathcal{U}[-1, 1]$. Initial velocity is set to zero. We evolve this setup to $T = 5$ and save 11 snapshots (every $\Delta t = 0.5$).

The **GCE-RT dataset** (Herde et al., 2024) likewise contains 1260 trajectories on a $128 \times 128$ grid, which we spatially downsample to $64 \times 64$. Each snapshot includes six fields ($\rho$, $v_x$, $v_y$, $p$, a tracer, and the gravitational potential $\phi$). In our setup, the passive tracer and gravitational potential are assumed available as inputs at every time step, while the model predicts only $[\rho, v_x, v_y, p]$. We follow the dataset split of Herde et al. (2024), using the first 1000 trajectories for training and the last 200 for validation. Since the raw data has 11 time frames, we take indices 0–3 as input and predict indices 4–10 (i.e. given the first 4 snapshots we predict the next 7).

## G Pseudocode of 2D SS-NO

```
class SS-NO(nn.Module):
    '''
    Notation:
        B: batch size
        T: temporal length
        X, Y: spatial dimensions
        C: input channels
        H: hidden dimension
    '''
```

```python
    def __init__(self,
                 lifting_layer: nn.Module,
                 projection_layer: nn.Module,
                 memory_pre_forward_x: nn.Module,
                 memory_pre_backward_x: nn.Module,
                 memory_pre_forward_y: nn.Module,
                 memory_pre_backward_y: nn.Module,
                 memory_t: nn.Module,
                 memory_post_forward_x: nn.Module,
                 memory_post_backward_x: nn.Module,
                 memory_post_forward_y: nn.Module,
                 memory_post_backward_y: nn.Module):
        super().__init__()
        self.p = lifting_layer
        self.q = projection_layer
        self.memory_pre_forward_x = memory_pre_forward_x
        self.memory_pre_backward_x = memory_pre_backward_x
        self.memory_pre_forward_y = memory_pre_forward_y
        self.memory_pre_backward_y = memory_pre_backward_y
        self.memory_t = memory_t
        self.memory_post_forward_x = memory_post_forward_x
        self.memory_post_backward_x = memory_post_backward_x
        self.memory_post_forward_y = memory_post_forward_y
        self.memory_post_backward_y = memory_post_backward_y

    def forward(self, x: Tensor) -> Tensor:
        '''
        Args:
            x: Input sequence of states (B, C, X, Y, T)
        Returns:
            Predicted next state (B, X, Y, 1)
        '''

        if self.training:
            x = x + torch.randn_like(x) * noise

        x = rearrange(x, 'b c x y t -> (b t) x y c')
        x = self.p(x)
        x = rearrange(x, '(b t) x y h -> (b t) h x y', t=T)

        # --- Pre-memory spatial SSM ---
        x = rearrange(x, '(b t) h x y -> (b t y) h x', t=T)
        x_fwd = self.memory_pre_forward_x(x)[0]
        x_bwd = torch.flip(self.memory_pre_backward_x(torch.flip(x, dims=[-1]))[0], dims=[-1])
        x = x_fwd + x_bwd
        x = rearrange(x, '(b t y) h x -> (b t) x h y', t=T)

        x = rearrange(x, '(b t) x h y -> (b t x) h y', t=T)
        y_fwd = self.memory_pre_forward_y(x)[0]
        y_bwd = torch.flip(self.memory_pre_backward_y(torch.flip(x, dims=[-1]))[0], dims=[-1])
        x = y_fwd + y_bwd
        x = rearrange(x, '(b t x) h y -> (b t) h x y', t=T)

        # --- Temporal SSM ---
        x = rearrange(x, '(b t) h x y -> b x y h t', t=T)
        x = rearrange(x, 'b x y h t -> (b x y) h t')
        x = self.memory_t(x)[0]
        x = rearrange(x, '(b x y) h t -> b x y h t', x=X, y=Y)
        x = rearrange(x, 'b x y h t -> (b t) h x y', t=T)

        # --- Post-memory spatial SSM ---
        x = rearrange(x, '(b t) h x y -> (b t y) h x', t=T)
        x_fwd = self.memory_post_forward_x(x)[0]
        x_bwd = torch.flip(self.memory_post_backward_x(torch.flip(x, dims=[-1]))[0], dims=[-1])
        x = x_fwd + x_bwd
        x = rearrange(x, '(b t y) h x -> (b t) x h y', t=T)

        x = rearrange(x, '(b t) x h y -> (b t x) h y', t=T)
```

```
        y_fwd = self.memory_post_forward_y(x)[0]
        y_bwd = torch.flip(self.memory_post_backward_y(torch.flip(x, dims=[-1]))[0], dims=[-1])
        x = y_fwd + y_bwd
        x = rearrange(x, '(b t x) h y -> (b t) h x y', t=T)

        x = self.q(x)
        x = rearrange(x, '(b t) c x y -> b x y t c', t=T)

        return x
```

Listing 3: SS-NO pseudocode

## H  Computational Efficiency and Cost-Accuracy Analysis

To rigorously evaluate the practical deployment viability of the proposed method, we conducted a comprehensive analysis of the trade-offs between predictive accuracy, theoretical complexity, and real-world inference latency.

### H.1  Benchmarking Methodology

All benchmarks were performed on a single NVIDIA A6000 GPU with a batch size of 1 to simulate real-time inference conditions. We report two key metrics across spatial resolutions $N \in \{32, 64, 128, 256, 512\}$:

- **Theoretical Complexity (GFLOPs):** The total number of floating-point operations (in billions) required for a single forward pass, calculated using the thop profiler with custom handlers for spectral operations.

- **Inference Latency (Time):** The wall-clock time in milliseconds, averaged over 100 iterations following a GPU warmup phase.

### H.2  Results and Discussion

The results are summarized in Table 6 and visualized as Pareto frontiers in Figure 12.

**State-Space Neural Operator (SS-NO).** Our model demonstrates the most favorable cost-accuracy profile. It achieves the lowest inference latency across all resolutions, consistently clocking under 0.85 ms. Notably, its runtime is effectively constant with respect to resolution in this regime, as the computation is dominated by fixed kernel launch overheads and the highly parallelizable associative scan of the S4 backbone.

**Galerkin Transformer (GKT).** We acknowledge the theoretical strength of the Galerkin Transformer, which exhibits the lowest GFLOP count of all models (0.03 GFLOPs at N=512). This efficiency stems from its linear attention mechanism, which avoids the quadratic complexity of standard Transformers. However, theoretical efficiency does not translate directly to wall-clock speed; GKT's inference latency ($\approx$2.26 ms) is nearly 3$\times$ higher than SS-NO due to the memory-bound nature of attention layers and the overhead of frequent reshaping operations.

**FFNO and U-Net.** The FFNO remains a strong competitor, offering low theoretical complexity (0.12 GFLOPs at N=512) and respectable runtime ($\approx$1.39 ms). However, SS-NO outperforms it in both speed ($\approx$1.7$\times$ faster) and accuracy. The U-Net and FactFormer architectures are significantly more expensive, with the U-Net requiring an order of magnitude more FLOPs (1.41 GFLOPs) for comparable or worse accuracy, highlighting the inefficiency of dense convolutional encoders for this class of problems.

In summary, while models like GKT and FFNO offer specific theoretical advantages, SS-NO achieves the optimal practical balance, delivering the highest accuracy with the lowest real-world latency.

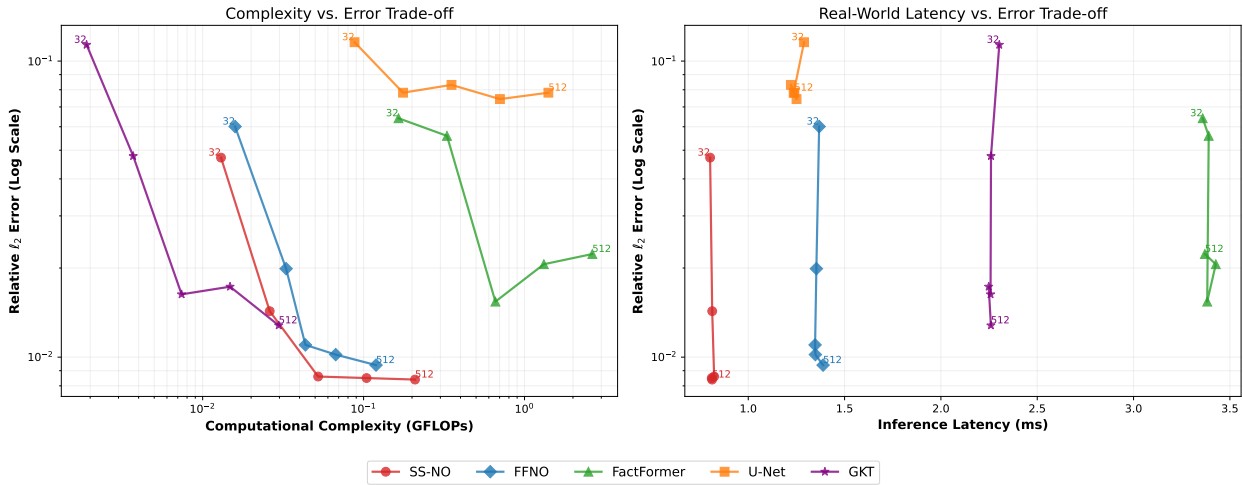

Figure 12: **Cost-Accuracy Trade-off Analysis (1D KS, $\nu = 0.075$).** We compare predictive error against computational cost across resolutions $N \in \{32, \ldots, 512\}$. **Left:** Theoretical Complexity (GFLOPs). SS-NO (Red) dominates the Pareto frontier, achieving the lowest error with significantly fewer operations than FactFormer and U-Net. **Right:** Empirical Inference Latency (ms). On an NVIDIA A6000 GPU, SS-NO is the fastest model ($< 0.85$ ms) and maintains constant runtime scaling. Note the discrepancy for GKT (Purple): despite low theoretical FLOPs, it suffers higher real-world latency due to attention overheads.

Table 6: **Computational Cost Summary.** Inference latency (ms) and theoretical complexity (GFLOPs) for all models across the full range of spatial resolutions ($N = 32$ to $512$). **SS-NO** consistently demonstrates the lowest inference latency ($\approx 0.81$ ms) and competitive scaling. While GKT shows very low theoretical FLOPs, its real-world latency is significantly higher due to attention overheads.

| Model | N=32 | | N=64 | | N=128 | | N=256 | | N=512 | |
|---|---|---|---|---|---|---|---|---|---|---|
| | Time (ms) | GFLOPs | Time (ms) | GFLOPs | Time (ms) | GFLOPs | Time (ms) | GFLOPs | Time (ms) | GFLOPs |
| U-Net | 1.29 | 0.088 | 1.24 | 0.176 | 1.22 | 0.352 | 1.25 | 0.704 | 1.24 | 1.407 |
| GKT | 2.30 | **0.002** | 2.26 | **0.004** | 2.26 | **0.007** | 2.25 | **0.015** | 2.26 | **0.030** |
| FactFormer | 3.36 | 0.165 | 3.39 | 0.330 | 3.38 | 0.659 | 3.43 | 1.319 | 3.37 | 2.638 |
| FFNO | 1.37 | 0.016 | 1.35 | 0.033 | 1.35 | 0.044 | 1.35 | 0.067 | 1.39 | 0.120 |
| **SS-NO (Ours)** | **0.80** | 0.013 | **0.81** | 0.026 | **0.82** | 0.052 | **0.81** | 0.104 | **0.81** | 0.209 |

# I  Formal Proof of FNO Recovery by SS-NO

In this section, we rigorously prove that the State Space Neural Operator (SS-NO) kernel formulation structurally subsumes the Fourier Neural Operator (FNO) kernel as a special case. Specifically, we show that by fixing the SS-NO parameters to specific values (zero damping, harmonic frequencies), the effective spatial convolution kernel of SS-NO becomes mathematically identical to the implicit spatial kernel of the FNO defined via the Discrete Fourier Transform (DFT).

## I.1  The FNO Kernel Formulation

The Fourier Neural Operator (FNO) operates by computing a convolution in the spectral domain. For a 1D input function $u(x)$ discretized on a grid of $N$ points $\{x_n = \frac{n}{N}\}_{n=0}^{N-1}$, the FNO computes the output $v$ as:

$$v = \mathcal{F}^{-1}(R \cdot \mathcal{F}(u))$$

where $R$ is a learnable weight matrix in the frequency domain. By the Convolution Theorem, this is equivalent to a circular spatial convolution $v = u * \kappa_{FNO}$, where the spatial kernel $\kappa_{FNO}$ is the IDFT of the spectral

weights $R$. Let $R_k$ denote the spectral weight matrix for the $k$-th frequency mode. The spatial kernel $\kappa_{FNO}$ evaluated at a grid point $x_n$ is:

$$\kappa_{FNO}[n] = \sum_{k=0}^{N-1} R_k e^{i\frac{2\pi k}{N}n} \quad \text{(Equation A)}$$

### I.2 The SS-NO Kernel Formulation

The SS-NO applies a convolution with a continuous kernel derived from the state equation. As detailed in Equation 4 of the main text, the effective spatial kernel is:

$$\kappa_{SSM}(x) = \sum_{m=1}^{M} C_m B_m^\top e^{-\rho_m |x|} e^{i\omega_m x}$$

Evaluating this on the discrete grid $x_n = \frac{n}{N}$ (assuming forward direction):

$$\kappa_{SSM}[n] = \sum_{m=1}^{M} (C_m B_m^\top) e^{-\rho_m \frac{n}{N}} e^{i\omega_m \frac{n}{N}} \quad \text{(Equation B)}$$

### I.3 Proof of Equivalence

We demonstrate that Equation B (SS-NO) can exactly recover Equation A (FNO) under the following restrictions: * Zero Damping ($\rho = 0$): Set $\rho_m = 0$ for all states, eliminating the decay term ($e^0 = 1$). * Harmonic Frequencies ($\omega = 2\pi k$): Set the continuous frequency parameter $\omega_m = 2\pi k$. On the grid, the exponent becomes $e^{i(2\pi k)\frac{n}{N}} = e^{i\frac{2\pi k}{N}n}$, recovering the DFT basis. * Spectral Weights (SVD Decomposition): Unlike FNO, which uses a full-rank matrix $R_k$ for channel mixing, a single SSM state provides a rank-1 mixing term $C_m B_m^\top$. However, any matrix $R_k$ can be decomposed into a sum of $J$ rank-1 terms (e.g., via Singular Value Decomposition): $R_k = \sum_{j=1}^{J} C_{k,j} B_{k,j}^\top$. To recover FNO, we set the number of SSM states to $M = N \times J$. We assign $J$ states to each frequency mode $k$, indexed by $j$. By initializing the parameters such that $\sum_{j=1}^{J} C_{k,j} B_{k,j}^\top = R_k$, Equation B becomes:

$$\kappa_{SSM}[n] = \sum_{k=0}^{N-1} \left( \sum_{j=1}^{J} C_{k,j} B_{k,j}^\top \right) e^0 e^{i\frac{2\pi k}{N}n} = \sum_{k=0}^{N-1} R_k e^{i\frac{2\pi k}{N}n}$$

**Conclusion.** This is mathematically identical to Equation A. Thus, the SS-NO formulation strictly subsumes the FNO kernel parametrization. SS-NO is a generalization because it allows for $\rho > 0$ (localized kernels), $\omega \neq 2\pi k$ (adaptive frequencies), and low-rank approximations where fewer than $J$ states are used per frequency.

## J Extensions to Irregular Geometries and Complex Unknown Systems

A primary motivation for Neural Operators is their ability to model systems where the governing PDEs are either **unknown, partially known, or computationally intractable** to solve directly. In such regimes, the operator must learn the underlying physics purely from observational data. In this section, we demonstrate SS-NO's capability in these challenging scenarios by extending it to irregular domains via coordinate transformation and slicing.

### J.1 Symmetry, Equivariance, and Geometric Inductive Bias

We first explicitly acknowledge a limitation raised by reviewers: the factorized scanning mechanism employed in our main text (processing dimensions sequentially, e.g., $x \to y$) is **not inherently equivariant** to rotation

or reflection. Unlike the isotropic kernels of standard CNNs or the global spectral bases of FNOs, the SSM scan introduces a directional inductive bias. For example, rotating a fluid field by 90° changes the order in which the SSM encounters features, potentially altering the output. Similarly, for periodic boundary conditions, the standard causal scan ($t \rightarrow t + 1$) breaks translation symmetry unless specific circular padding schemes are used.

To reconcile this, we explore combining SS-NO with existing coordinate transformation frameworks, specifically utilizing the **diffeomorphic mappings** introduced in Geo-FNO (Li et al., 2023b). By decoupling the scanning grid from the physical domain, we can maintain the computational efficiency of the factorized scan while respecting the geometric properties of the data.

### J.2 Approach 1: Recovering Symmetry via Transformations (Shallow Water Equations)

To validate SS-NO on a system with complex topology and no explicit PDE supervision, we test on the spherical Shallow Water Equations (SWE).

To address the symmetry breaking discussed above, we adopt the coordinate transformation approach from Geo-FNO (Li et al., 2023b). This method learns a **diffeomorphism** that maps the physical manifold (the sphere) to a uniform computational latent grid. Crucially, because this mapping is diffeomorphic, it can preserve spatial symmetries—such as rotation and reflection—in the physical space, even if the internal computational scan remains sequential. The transformation layer effectively "unwraps" the domain into a canonical representation where the directional bias of the SSM is mitigated.

Experimental results against the Spherical FNO (SFNO) suggest that SS-NO's *adaptive frequency learning* is uniquely suited for this transformed space. In a mapped domain, physical waves often manifest as warped, non-harmonic oscillations that fixed Fourier bases struggle to represent sparsely. SS-NO automatically adapts its continuous frequency parameters $\omega_m$ to these distorted geometries, effectively learning a physics-compliant basis from data.

To rigorously test whether this combination has learned the underlying operator, we utilized a strict evaluation protocol: models were trained on a short horizon (10 steps) but evaluated on stability during long rollouts up to $t = 100$. As shown in Table 7, SS-NO demonstrates superior stability, achieving $\sim 2.2\times$ lower error than the baseline at $t = 100$, confirming that the learned diffeomorphism successfully preserved the necessary global dynamics.

### J.3 Approach 2: Complex Aerodynamics via Slicing (AirfRANS)

We further evaluate SS-NO on the **AirfRANS** dataset (Bonnet et al., 2022), representing a "partially known" or complex system where analytic solutions are unavailable. This task involves predicting velocity, pressure, and skin friction fields over NACA airfoils. Here, the "true" PDE (Navier-Stokes) is computationally prohibitive for design optimization, and the model must act as a surrogate learned entirely from RANS simulation data.

For this irregular mesh problem, we adopt the slicing-based approach from Transolver (Wu et al., 2024). We integrated the SS-NO block into the Transolver architecture, replacing its core attention mechanism. As shown in Table 8, SS-NO achieves competitive regression performance, significantly lowering **Volume Error** (0.0017 vs 0.0037) and **Surface Error** (0.0046 vs 0.0142) compared to the baseline.

However, the model lags slightly in derived physical coefficients (Lift $C_L$). We hypothesize this is due to the Linear Time-Invariant (LTI) nature of the standard S4 backbone, which applies uniform dynamics globally. Complex aerodynamics require spatially varying dynamics (e.g., distinguishing laminar leading edges from turbulent wakes). We anticipate that replacing the S4 backbone with input-dependent State Space Models (e.g., **Mamba** (Gu & Dao, 2024)), which can dynamically modulate parameters based on the local flow state, would bridge this gap.

Table 7: **Spherical Shallow Water Equations (SWE) Benchmark.** Comparison of relative $\ell_2$ errors on spherical SWE forecasting over long time horizons. Models were trained on a short horizon (10 steps) and evaluated up to $t = 100$. SS-NO demonstrates superior stability and generalization, achieving $\sim 2.2\times$ lower error than the Spherical FNO (SFNO) at $t = 100$, indicating it has effectively learned the global atmospheric dynamics from data.

| Evaluation Step | SFNO (Baseline) | SS-NO (Ours) | Improvement |
|---|---|---|---|
| $t = 20$ | $6.56 \times 10^{-2}$ | $\mathbf{2.63 \times 10^{-2}}$ | $2.5\times$ |
| $t = 50$ | $5.99 \times 10^{-1}$ | $\mathbf{2.54 \times 10^{-1}}$ | $2.4\times$ |
| $t = 100$ | $2.82 \times 10^{0}$ | $\mathbf{1.28 \times 10^{0}}$ | $2.2\times$ |

Table 8: **AirfRANS Benchmark (Data-Driven Aerodynamics).** Comparison of SS-NO against Transolver on the AirfRANS dataset. This task represents a "partially known" system where the model must learn complex aerodynamic mappings purely from data. SS-NO excels at field reconstruction (Volume/Surface error) but requires input-dependent gating (e.g., Mamba) to better capture global coefficients like Lift.

| Model | Error Vol. ($\downarrow$) | Error Surf. ($\downarrow$) | Lift Coef. $C_L$ ($\downarrow$) | Spearman $\rho_L$ ($\uparrow$) |
|---|---|---|---|---|
| Transolver | 0.0037 | 0.0142 | **0.1030** | **0.9978** |
| **SS-NO (Ours)** | **0.0017** | **0.0046** | 0.1098 | 0.9973 |

# K  Background: The MemNO Framework

In this work, we adopt the training and inference strategy proposed in the **Memory Augmented Neural Operator (MemNO)** framework Buitrago et al. (2025). As this framework serves as the temporal backbone for our experiments, we provide a summary of its theoretical motivation and architectural design here.

## K.1  Theoretical Motivation: Mori-Zwanzig and Coarse-Graining

The primary motivation for MemNO stems from the **Mori-Zwanzig formalism** in statistical mechanics. This theory addresses the problem of modeling a dynamical system when the full state is not observable—specifically, when the system is projected onto a lower-dimensional or coarser representation (e.g., discretizing a PDE onto a coarse spatial grid).

Even if the underlying PDE (the microscopic dynamics) is Markovian—meaning the future depends only on the current exact state—the *coarse-grained* dynamics are generally **non-Markovian**. The resolved (observable) scales interact with the unresolved (sub-grid) scales, and the influence of these unresolved scales manifests as a "memory" of the system's history. Mathematically, the evolution of the coarse variables depends on an integral over the past states (the memory kernel).

Standard autoregressive Neural Operators often ignore this, treating the coarse mapping as Markovian ($\hat{u}_{t+1} = \mathcal{G}(u_t)$). MemNO argues that to accurately model coarse-grained PDE trajectories, the architecture must explicitly account for this memory term to compensate for the loss of information due to spatial discretization.

## K.2  Architectural Implementation: S4 Temporal Layers

MemNO employs S4 layers along the temporal dimension. The temporal S4 mechanism allows the model to efficiently compress the entire history of the input function into a compact latent state. This enables the model to effectively learn the non-Markovian memory kernel required to close the system dynamics, essentially using temporal information to recover spatial details lost during coarse-graining.

### K.3 Adoption in This Work

We adopted the MemNO framework for our temporal benchmarks for two key reasons:

1. **Handling Coarse Resolutions:** Many of our benchmarks (e.g., Navier-Stokes at $64 \times 64$) operate in regimes where the spatial grid does not fully resolve the turbulence. The MemNO framework allows all models to utilize temporal history to compensate for this aliasing, ensuring a physically rigorous evaluation.

2. **Standardized Temporal Backbone:** By fixing the temporal architecture to use MemNO's S4-based time mixing for all baselines (SS-NO, FNO, FactFormer, etc.), we ensure a fair comparison. This isolates the contribution of the *spatial* mixing architectures, confirming that the performance gains reported for SS-NO arise from its superior spatial processing rather than an advantageous temporal integrator.

## L   Limitations

Despite the significant reduction in model size and strong performance achieved by SS-NO, certain limitations remain inherent to the current formulation.

First, the **accumulation of gradients** during autoregressive training poses a scalability bottleneck. While SS-NO is parameter-efficient, increasing the state dimension $D$ or the sequence length $L$ linearly increases the memory required for backpropagation through time. This restricts the feasibility of adopting extremely large latent states without specialized techniques. However, unlike language modeling where infinite-horizon dependencies are crucial, PDE dynamics are often effectively Markovian, suggesting that techniques such as truncated backpropagation or gradient checkpointing could provide effective mitigation.

Second, the **sequential scanning mechanism** introduces a directional inductive bias that is not inherently equivariant. Unlike isotropic convolutions or global spectral layers, the state-space scan processes data sequentially (e.g., $x \rightarrow y$), which breaks rotational and reflection symmetries. While we have shown in Appendix J that this can be managed via symmetric embedding layers or coordinate transformations, the core mixing block itself does not guarantee equivariance by construction.

## M   Possible Extensions and Future Work

We see several promising directions to extend the SS-NO framework, particularly in enhancing its expressivity for complex dynamics and further generalizing its geometric capabilities.

**Input-Dependent Backbones (Mamba and Hydra).**   A distinct advantage of the SS-NO framework is its modularity; the core scanning backbone can be swapped for more advanced architectures. While the current implementation utilizes efficient Linear Time-Invariant (LTI) systems, a natural extension is to incorporate **input-dependent backbones** such as Mamba (Gu & Dao, 2024) or Hydra (Hwang et al., 2024). These models allow state transition matrices to be functions of the input field, enabling the operator to dynamically modulate its effective receptive field and frequency response based on local flow features (e.g., shock waves or boundary layers). This would be particularly beneficial for multi-regime systems like the AirfRANS benchmark discussed in Appendix J.

**Native Graph and Manifold Learning.**   While we successfully demonstrated extending SS-NO to irregular domains via slicing and coordinate transformation, these methods essentially project data onto regular latent structures. A more fundamental approach is to generalize the scanning operator to work *natively* on graphs or meshes. Recent works like Graph Mamba (Wang et al., 2024) and pLSTM (Pöppel et al., 2025) define propagation orders over graph topologies. Adopting these backbones would allow SS-NO to process unstructured meshes without the need for interpolation or slicing, offering a principled way to handle complex engineering geometries.

**True Geometric Recurrence.** A more theoretical direction is to redefine the recurrence relation itself using geometric differential operators. This would entail replacing the ordinary time derivative in the SSM with a **covariant derivative**,

$$\nabla_{\dot{\gamma}(t)} h = Ah + Bu,$$

incorporating notions of parallel transport to evolve the hidden state along geodesics. Such an extension would enable genuinely Riemannian state evolution, preserving vector field symmetries on curved manifolds by construction, though it introduces significant mathematical complexity.

**Memory-Efficient Training Strategies.** Finally, improving the memory efficiency of autoregressive training remains a priority. Given that turbulent flows often exhibit chaotic but bounded attractors, exact gradient retention over long horizons may yield diminishing returns. We plan to investigate **truncated backpropagation** and **implicit differentiation** strategies specifically optimized for Neural Operators, aiming to balance the fidelity of long-term gradients with the computational constraints of high-resolution 3D simulations.

