# OpenReview forum: "Merging Memory and Space: A State Space Neural Operator"
_TMLR — Accepted by TMLR_

### Review · Reviewer_JDKn · 2025-10-25

**Summary Of Contributions:**

The paper introduces the State Space Neural Operator (SS-NO), a compact and efficient neural architecture that unifies spatiotemporal operator learning for time-dependent partial differential equations (PDEs) using structured state space models. The SS-NO model merges insights from neural operators and structured state space models (SSMs) to learn mappings between function spaces defined by PDEs. Traditional models like the Fourier Neural Operator (FNO) capture global dependencies but suffer from high memory costs, while state space models like S4 efficiently model long-range temporal patterns. SS-NO generalizes FNO by incorporating adaptive damping (which localizes the receptive field for stability) and learnable frequency modulation (which selects spectral modes dynamically), enabling efficient and expressive spatiotemporal modeling. Architecturally, SS-NO employs bidirectional 1D spatial SSMs along each axis and integrates a temporal memory module following the MemNO framework. A theoretical universality theorem is proven, showing that convolutional neural operators with full field-of-view—including SS-NO—can approximate any continuous operator. The model is evaluated on several 1D and 2D PDE benchmarks, including Burgers’, Kuramoto–Sivashinsky, Navier–Stokes, and compressible Euler equations. SS-NO outperforms existing methods in accuracy while using significantly fewer parameters. Extensive ablation studies analyze the role of damping, frequency modulation, temporal memory, and architectural variants, revealing that SS-NO’s adaptive mechanisms are critical for performance, especially in chaotic or low-resolution regimes.

**Additional Comments:**

The paper is well written, so I have no additional comments, except for a minor typo:
The paper used both 'field-of-view' and 'field of view'. Choose one form consistently.

**Audience:**

Yes

**Audience Explanation:**

Many researchers in the TMLR audience would find modeling state space models for time-dependent partial differential equations (PDEs) useful.

**Claims And Evidence:**

Yes

**Claims Explanation:**

SS-NO is reportedin the paper to consistently achieve state-of-the-art accuracy across 1D and 2D PDE benchmarks while reducing parameter count by up to 180× compared to baselines. It excels particularly in chaotic or low-resolution scenarios, where adaptive damping and spectral modulation enhance stability and generalization, validating both its theoretical expressivity and empirical robustness. The paper provides a sharp universality criterion for convolutional neural operators with full field-of-view, filling a theoretical gap in the operator learning literature. The paper also carefully isolates the impact of each architectural component (damping, frequency learning, temporal memory) across multiple regimes and capacities. Finally, the benchmarks include diverse dynamical systems with varying dimensionalities, physics regimes, and complexities, making the conclusions broadly relevant.
The paper also has some weaknesses. The method assumes regular spatiotemporal grids, with no extensions to unstructured meshes or irregular geometries. Also, training remains memory-intensive due to backpropagation through long sequences, especially when increasing state dimensionality. Finally, while adaptivity to missing context is shown, the role of SS-NO under real-world noisy or multimodal inputs (e.g., from sensors or images) could be further clarified.

**Requested Changes:**

I would suggest the authors to adress some of the comments related to how the paper could be improved above. One specific suggestion concerns extending SS-NO to irregular domains or unstructured meshes. Is it possible to reformulate SS-NO on graph domains using graph-based state-space models (e.g., graph SSMs or message-passing variants of S4). Alternatively, the analysis could be applied to extend the universality theorem by defining “full field-of-view” over geodesic or graph distances on manifolds or meshes, enabling theoretical guarantees beyond Euclidean grids.. This would allow SS-NO to operate over unstructured spatial inputs. If so, that could strengthen the model to irregular domains or unstructured scenarios. If a theoretical approach is not possible an empirical meta-analysis could help. For instance, evaluate SS-NO variants on PDEs defined over triangulated meshes (e.g., FEM or CFD simulations on irregular domains), or combine SS-NO with graph neural operators (e.g., MGNO, Diffusion GNNs) and test interpolation/generalization on mesh-refinement or boundary deformation tasks.

---

> ### Author Response · Authors · 2026-01-27
> **Response to Reviewer JDKn**
>
> We sincerely thank the reviewer for their thorough reading and highly positive assessment of our work. We appreciate the accurate summary of our contributions and are encouraged by the recognition of SS-NO’s performance in chaotic regimes and its theoretical "full field-of-view" contribution.
>
> We have addressed the requested changes regarding irregular domains and typos as follows:
>
> **1. Extensions to Irregular Domains and Unstructured Meshes**
>
> The reviewer suggested extending SS-NO to irregular domains, either theoretically (via graph SSMs) or empirically (via evaluation on meshes). We agree this is a critical direction. To address this, we have added a comprehensive new section, **Appendix J: Extensions to Irregular Geometries and Complex Unknown Systems**, where we go beyond theoretical discussion and provide concrete empirical evidence using two distinct strategies:
>
> * **Coordinate Transformation (Diffeomorphic Mapping):** For domains with complex topology but continuous structure (like the sphere), we combined SS-NO with a learnable coordinate transformation (similar to Geo-FNO). We validated this on the Spherical Shallow Water Equations (SWE). The results (Table 7) show that SS-NO adapts well to the transformed grid, outperforming the baseline Spherical FNO by  on long-term stability tests.
> * **Domain Slicing:** For true unstructured meshes, we integrated SS-NO into a slicing-based framework (following the Transolver architecture) and evaluated it on the AirfRANS dataset (NACA airfoil optimization). SS-NO demonstrated strong capabilities in field reconstruction, significantly reducing regression errors compared to the baseline. However, we also candidly report that it performed slightly worse on physics-derived metrics like the Lift Coefficient (Table 8), likely because the current linear backbone applies uniform dynamics globally. This points to a clear future direction: utilizing input-dependent backbones (e.g., Mamba) to better capture spatially varying flow regimes (e.g., laminar vs. turbulent).
>
> In addition to these empirical results, we have updated the **"Possible Extensions and Future Work"** section (Appendix K) to explicitly discuss the theoretical path toward **native graph-based SSMs** (e.g., Graph Mamba, pLSTM) and **geometric recurrence** using covariant derivatives, as suggested by the reviewer.
>
> **2. Memory Complexity and Gradient Management**
>
> We acknowledge the reviewer's comment regarding the memory intensity of backpropagation through long sequences. In the revised **Limitations** section (Appendix L), we have expanded the discussion on this tradeoff. We note that for PDE dynamics, which are often effectively Markovian, techniques such as truncated backpropagation or gradient checkpointing offer a viable path to mitigate these costs without sacrificing accuracy.
>
> **3. Typos**
>
> We have corrected the inconsistency regarding "field-of-view" vs "field of view" throughout the manuscript. We consistently use "field-of-view" in adjectival phrases and "field of view" as a noun phrase.
>
> We believe these additions—particularly the new experimental evidence on Spherical SWE and AirfRANS—directly address the reviewer's request to demonstrate SS-NO's applicability to irregular and unstructured scenarios.

---

### Review · Reviewer_Dod9 · 2025-12-07

**Summary Of Contributions:**

This paper introduces the State Space Neural Operator (SS-NO), which extends SSMs to spatiotemporal operator learning for time-dependent PDEs. The main contributions are: 1. a bidirectional spatial SSM architecture that applies S4D-like convolutions along each spatial dimension sequentially, 2. a learnable damping coefficients and frequency modulation that provide adaptive receptive fields, and 3. a universality theorem showing that convolutional neural operators with "full field of view" are universal approximators.

**strengths**
1. The parameter efficiency is well notable.
2. The ablation study on damping mechanisms is thorough and provides good insight into when/why damping helps
3. The theoretical connection between SSMs and FNOs through the kernel formulation (Eq. 4-5) is interesting

**weaknesses**
1. The claim that SS-NO "subsumes" F-FNO is not clearly demonstrated
2. The universality proof, while technically sound, has unclear practical implications
3. Breaking of spatial symmetries (rotation, reflection, translation) is not discussed
4. Benchmarks are relatively easy and do not test settings where neural operators are most useful (PDEs not known)

**Audience:**

Yes

**Audience Explanation:**

The neural operator community may find this work relevant, particularly those interested in efficient architectures for PDE learning. The connection between SSMs and neural operators is timely given the recent success of SSMs in sequence modeling. The practical parameter efficiency gains are meaningful for deployment scenarios. The damping analysis also provides useful insights that could inform future architecture design.

**Claims And Evidence:**

No

**Claims Explanation:**

The experimental results are in general well-presented and the ablation studies are convincing for the claims about damping. However, several theoretical claims are not adequately supported:

1. The paper claims SS-NO "subsumes" F-FNO as a special case, but the actual mechanism for recovering F-FNO kernels is hand-wavy. Setting $\rho_k = 0$ and $\omega_k = k$ in Eq. 5 does not obviously recover the F-FNO parameterization since the kernel structure differs (exponential decay vs truncated Fourier series).

2. The universality theorem (Theorem 4.1) is proven rigorously in the appendix, but its purpose is unclear. Both FNO and SS-NO trivially satisfy the full field-of-view condition, so what does this result add? The paper states it "closes a gap" for factorized architectures, but doesn't explain why this gap mattered or what practical insight we gain.

3. The claim about "enhanced model expressivity" over F-FNO (Section 4.2) is theoretical but the empirical comparison doesn't isolate this effect. SS-NO also has different architecture choices beyond the kernel parameterization.

**Requested Changes:**

**critical**
1. Clarify or rephrase the claim that SS-NO subsumes F-FNO. Currently it reads as if one can exactly recover F-FNO, but the kernel parameterizations are clearly different. Either provide a rigorous proof of this subsumption or reframe as "inspired by" or "related to."
2. Add discussion on spatial symmetry breaking. The sequential application of 1D SSMs along x then y (and forward and backward) breaks rotational and reflection symmetries (and also translation symmetries for periodic boundaries). For many physical systems these are important inductive biases. The authors should discuss: (a) how this affects results on rotationally/reflectively/translationally symmetric problems, (b) whether data augmentation helps, (c) comparison to architectures that preserve symmetry.
3. Clarify the purpose of the universality theorem. As stated, it's not clear what new insight this provides beyond confirming that both FNO and SS-NO are universal. If the main point is about factorized architectures, please explain why this was previously uncertain.
4. Include experiments on problems with unknown or partially known PDEs. This is arguably the most important use case for neural operators. Showing that SS-NO works when we only have data (no explicit PDE) would strengthen the paper considerably.
5. Test on more challenging 2D/3D problems. The 1D benchmarks dominate the paper but are relatively simple. Higher-dimensional turbulence or multi-physics problems would be more convincing.
6. Fix grammatical issues, e.g., "the formulation equation 6" should be "the formulation in Equation 6" or similar. Also check consistency in notation.

**strengthen**
1. The MemNO framework is referenced repeatedly but the connection could be explained more clearly for readers unfamiliar with that work.
2. Some baseline comparisons may be unfair. FNO2D with 67M parameters is clearly overparameterized for these tasks. A capacity-matched comparison would be informative.

---

> ### Author Response · Authors · 2026-01-27
> **Response to Reviewer Dod9**
>
> We thank the reviewer for their thoughtful feedback and for recognizing the parameter efficiency of our method and the value of our damping ablation study. We have significantly updated the manuscript to address your concerns regarding benchmark difficulty, theoretical claims, and symmetry. Specifically, we have added **three new appendices (I, J, K)** containing rigorous proofs, new experiments on irregular domains, and background on the MemNO framework.
>
> Below is our point-by-point response.
>
> **1. Response to "Benchmarks are relatively easy" & "Unknown PDEs"**
>
> We first clarify that we tested on **five distinct 2D PDE benchmarks** (Navier-Stokes, Compressible Euler, etc.). The extensive use of 1D benchmarks in the main text was primarily for rigorous ablation studies and visualization of the damping mechanism, rather than to simplify the task.
>
> Regarding the concern about "unknown PDEs," we interpreted this in two ways and conducted **two new 2D experiments on irregular domains** to address both:
>
> * **Interpretation A: Dynamics unseen during training.** We tested on the **2D Spherical Shallow Water Equations (SWE)**. While the equations are known to the simulator, the model observes only a short time horizon (10 steps) and must generalize to a long horizon ($t=100$) where the dynamics evolve significantly.
> * *Results:* SS-NO demonstrates superior stability, achieving $\approx 2.2 \times$ lower error than the SFNO baseline at $t=100$ ($1.28$ vs $2.82$).
>
>
> * **Interpretation B: Partially known systems/Surrogates.** We evaluated SS-NO on the **2D AirfRANS** dataset (NACA airfoil optimization), where the model acts as a surrogate for complex RANS simulations on irregular meshes. We utilized a slicing-based approach (integrated into Transolver).
> * *Results:* SS-NO achieved competitive regression performance, significantly lowering **Volume Error** ($0.0017$ vs $0.0037$) compared to the baseline.
> * *Trade-off:* We candidly note that while regression accuracy improved, SS-NO yielded a slightly higher error on the physics-derived **Lift Coefficient** ($C_L$) and a lower **Spearman correlation**. We attribute this to the global LTI nature of the current backbone, suggesting a need for input-dependent gating (like Mamba) for multi-regime flows.
>
>
>
> **2. Response to "Spatial Symmetry Breaking"**
>
> We explicitly acknowledge the limitation raised by the reviewer: the factorized scanning mechanism employed in our main text (processing dimensions sequentially, e.g., $x \rightarrow y$) is not inherently equivariant to rotation or reflection. Unlike the isotropic kernels of standard CNNs or global spectral bases, the SSM scan introduces a directional inductive bias. For example, rotating a fluid field by $90\degree$ changes the order in which the SSM encounters features.
>
> To reconcile this, we have added **Appendix J.1**, where we explore combining SS-NO with existing coordinate transformation frameworks, specifically utilizing **diffeomorphic mappings** (as in Geo-FNO). By decoupling the scanning grid from the physical domain, we can maintain the computational efficiency of the factorized scan while respecting the geometric properties of the data. This approach was successfully validated in the Spherical SWE experiment mentioned above.
>
> **3. Response to "Subsumption of F-FNO"**
>
> We acknowledge the imprecision in our original text regarding the subsumption claim. We have revised the manuscript to be precise and have added **Appendix I**, which contains a formal proof showing that the SS-NO kernel parametrization structurally subsumes the FNO kernel under specific restrictions. We invite the reviewer to examine Appendix I for the full derivation.
>
> **4. Response to "Universality Theorem Purpose"**
>
> The reviewer asked what practical insight the universality theorem adds.
> As far as we are aware, there are **no available proofs** for the universality of any factorized architectures. The standard proof for FNO relies on global integration and complete bases in every layer.
> Theorem 4.1 fills this gap by identifying the minimal condition required for universality: a **"full field-of-view."** The theorem implies that it is essentially impossible to create a non-universal neural operator architecture—even with factorized or non-tunable kernels—provided the receptive field covers the domain. This provides the first theoretical footing for the entire class of factorized neural operators.
>
> **5. Baseline Fairness and Hyperparameter Tuning**
>
> To ensure fairness, we conducted a thorough hyperparameter search for spectral baselines. On `TorusLi`, testing  modes revealed that 8 modes significantly outperformed higher settings, and we updated the results accordingly. For other benchmarks, standard configurations remained optimal. We hypothesize this disparity stems from the non-constant viscosity and forcing in `TorusVis` and `TorusVisForce`, where higher mode counts are necessary to effectively ``multi-task'' across varied physical regimes.

---

> ### Author Response · Authors · 2026-01-27
> **MemNO Background and Grammatical Issues**
>
> Finally, to address the request for clarity regarding our temporal backbone, we have added Appendix K, providing a dedicated background on the MemNO framework to explain the memory mechanism for readers unfamiliar with that work. Additionally, we have carefully proofread the manuscript and corrected the grammatical issues pointed out by the reviewer (e.g., consistency in notation and equation referencing).

---

### Review · Reviewer_V25i · 2026-01-21

**Summary Of Contributions:**

The main contributions of this work are:
1. Connects state space models to neural operators, and in particular, FNOs. It is shown that in 1d a SSM can exactly recover a FNO that can also modulate the frequency components, giving it the ability to more easily localize.
2. Proposes a factorized SSM by applying a 1d SSM in each direction similarly to a factorized transform like the FFT.
3. Proves a general universal approximation theorem for integral operators of the convolution type, showing that all that is needed is for the composition of all the kernels in each layer to have global support.
4. Extensive numerical experiments on Burger's equation, the Kuramoto–Sivashinsky equation, and the Navier-Stokes equation show the method is promising and outperforms 4 to 5 baselines.

**Audience:**

Yes

**Audience Explanation:**

The paper is significant for the field of data-driven surrogates models for PDEs, connecting promising methods in NLP to operator learning.  This is a significant audience for TMLR.

**Broader Impact Concerns:**

There are no ethical implications.

**Claims And Evidence:**

Yes

**Claims Explanation:**

The theory is well-formulated and the poofs given in the appendix are fully rigorous. The numerical experiments are conducted over 5 random initialization of the model weights and show statistically significant improvements.

**Requested Changes:**

1. In 1d, the resolution dependence of all the models can be tested more rigorously. The maximum resolution of 128 is quite small and doesn't demonstrate the resolution dependence of the U-net likely because the architecture tested still has a fixed field of view at that resolution. It would be helpful to put these results on a plot instead of in tables as it makes comparisons much easier.
2. The run-time cost of any of the models is not consider. Parameter count is not really good proxy for this. Cost-accuracy trade-off curves would be very useful, especially as the resolution grows (in 2d and in 1d). This can be done w.r.t. a theoretical calculation for the number of flops for each method or the run-time on the same hardware (or both).
3. The authors claim in the text that the kernel having a "full field of view" is a necessary and sufficient condition for universality but the theorem is stated as the condition only being sufficient. Could the authors clarify and provide extra discussion if condition is indeed not necessary.

---

> ### Author Response · Authors · 2026-01-27
> **Response to Reviewer V25i**
>
> We sincerely thank the reviewer for their positive assessment and for recognizing the significance of connecting NLP-style State Space Models to Operator Learning. We appreciate the praise regarding the rigor of our proofs and the breadth of our numerical experiments.
>
> We have addressed the requested changes as follows:
>
> **1. Extended 1D Resolution Study (Plots vs. Tables)**
>
> Following the reviewer's suggestion to test resolution dependence more rigorously, we have significantly expanded our evaluation on the 1D Kuramoto-Sivashinsky equation to include higher spatial resolutions of $N = 256$ and $N = 512$ (previously capped at 128). We have visualized these results in **Figure 3** to facilitate easier comparison.
>
> Our analysis reveals a distinct contrast between explicit spectral methods and our state-space approach:
>
> * **Explicit Spectral Scaling (FFNO):** The FFNO shows a slight but continuous reduction in error at higher resolutions. We attribute this to the explicit nature of the FFT; as the grid resolution increases, the Nyquist frequency rises, allowing FFNO to capture high-frequency details that were previously truncated.
> * **Implicit Adaptive Learning (SS-NO):** In contrast, SS-NO achieves its optimal performance floor earlier (effectively converging at $N = 128$) and maintains this superior accuracy at $N = 512$. This supports our hypothesis regarding the **adaptive frequency learning** of SSMs. Unlike FFNO, which requires finer spatial discretization to resolve high frequencies, SS-NO’s adaptive mechanism extracts these dynamics efficiently within its latent state space.
>
> We note that this extended resolution study was conducted specifically on the 1D Kuramoto-Sivashinsky equation. The available ground-truth simulation data for the other benchmarks is limited: the 1D Burgers' equation dataset is capped at $N=128$, and the 2D benchmarks (e.g., Navier-Stokes, Compressible Euler) are provided at maximum resolutions of $64 \times 64$.
>
> **2. Cost-Accuracy Trade-off Analysis**
>
> We agree that parameter count is an imperfect proxy for cost. Following part 1's results, we have added a comprehensive **Computational Efficiency analysis in Appendix H**, reporting both Theoretical Complexity (GFLOPs) and real-world Inference Latency (Wall-clock time on an NVIDIA A6000 GPU).
>
> * **Pareto Optimality:** As visualized in the new Pareto frontiers (Figure 12), SS-NO demonstrates the most favorable profile. It achieves the lowest inference latency (< 0.85 ms) across all resolutions.
> * **Theoretical vs. Practical Cost:** We highlight a key discrepancy for the Galerkin Transformer (GKT). While GKT exhibits the lowest theoretical complexity (0.03 GFLOPs), its real-world latency ($\approx$ 2.26 ms) is nearly 3$\times$ **slower** than SS-NO due to the memory-bound nature of attention and reshaping overheads.
> * **Comparison to FFNO:** While FFNO is efficient ($\approx$ 1.39 ms), SS-NO outperforms it in both speed ($\approx$ 1.7$\times$ faster) and accuracy. Denser models like U-Net are shown to be significantly more expensive ($\approx$ 1.41 GFLOPs) for comparable accuracy.
>
> **3. Universality: Necessity vs. Sufficiency**
>
> We thank the reviewer for their keen eye on this theoretical point. Indeed, our theorem strictly proves only the **sufficiency** of the full field-of-view property. To clarify this, we have removed references to it being a "necessary condition" in the main text.
>
> However, we believe this condition is intuitively "minimal." We have added **Appendix B.2** to provide further discussion on this:
>
> * **Subtleties:** Rigorously proving necessity is complicated by potential cancellations. For example, if composing kernels $\kappa_{\ell}$ on a torus results in disjoint spatial Fourier modes, their convolution could equate to zero even if the architecture is expressive, technically violating the field-of-view requirement while potentially retaining universality.
> * **Conjecture:** We conjecture that the full field-of-view property becomes necessary under specific conditions—for instance, if the architecture is restricted to non-negative kernels ($g_{\ell} \geq 0$), or if the architecture satisfies a property where representability of a function $g_{\ell}$ implies representability of $\|g_{\ell}\|$.
>
> We believe these revisions make the theoretical claims fully accurate while providing the rigorous cost and resolution analysis requested.

---

### Author Response · Authors · 2026-02-06
**Thank you to Reviewers and AE**

We thank the reviewers for their detailed engagement and constructive suggestions, which have greatly improved the quality of this work. We hope that the additional experiments and theoretical proofs provided in the revision serve to fully address the points raised. We also thank the Action Editor for their time and effort in managing the review process.

---

### Decision · Action_Editor_X9uL · 2026-03-03

**Recommendation:** Accept as is

**Audience:**

Yes

**Audience Explanation:**

The reviewers are all leaning towards acceptance and find the paper relevant to the TMLR audience, particularly in the context of learning data-driven surrogate models for PDEs. The correspondence between SSM and NOs might be interesting to the broader audience.

**Claims And Evidence:**

Yes

**Claims Explanation:**

The reviewers unanimously recognize the correctness of the paper.